# Suspected Pituitary Apoplexy: Clinical Presentation, Diagnostic Imaging Findings and Outcome in 19 Dogs

**DOI:** 10.3390/vetsci9040191

**Published:** 2022-04-15

**Authors:** Greta Galli, Giovanna Bertolini, Giulia Dalla Serra, Marika Menchetti

**Affiliations:** 1Neurology and Neurosurgery Division, San Marco Veterinary Clinic, 35030 Veggiano, Italy; marika.menchetti@sanmarcovet.it; 2Diagnostic and Interventional Radiology Division, San Marco Veterinary Clinic, 35030 Veggiano, Italy; giovanna.bertolini@sanmarcovet.it (G.B.); giulia.dallaserra@sanmarcovet.it (G.D.S.)

**Keywords:** pituitary apoplexy, pituitary mass, neurological examination, dog, magnetic resonance imaging, computed tomography

## Abstract

In human medicine, pituitary apoplexy (PA) is a clinical syndrome characterised by the sudden onset of neurological signs because of haemorrhage or infarction occurring within a normal or tumoral pituitary gland. The diagnosis is usually performed combining neurological signs and imaging findings. The aim of the present study is to describe the abnormal neurological signs, the diagnostic imaging findings, based on Computed Tomography (CT) and/or Magnetic Resonance Imaging (MRI), and the outcome in a population of dogs with suspected PA. Clinical cases were retrospectively reviewed. Nineteen cases of suspected PA were included. The majority of dogs showed behavioural abnormalities (11/19). Neurological signs more frequently identified were obtundation (7/19), vestibular signs (7/19) and epileptic seizures (6/19). The onset of neurological signs was per-acute in 14 out of 19 cases. Data regarding CT and MRI were available in 18 and 9 cases, respectively. Neurological signs resolved in less than 24 h in seven patients. The short-term prognosis was defined as favourable in the majority of our study population. The median survival time was of 7 months from the time of PA diagnosis. This is the first description of neurological signs, imaging findings and outcome in a large group of dogs with PA.

## 1. Introduction

Pituitary apoplexy (PA) is defined as a haemorrhage or infarction of the pituitary gland, resulting in the sudden onset of neurological signs [1,2,3,4]. In human medicine, PA usually occurs within a pre-existing pituitary tumour, but can also involve the normal pituitary tissue [1,2,3,4]. However, to the authors’ knowledge, a description of PA occurring in a normal pituitary gland is lacking in the veterinary literature. In human medicine, the majority of patients are unaware of the pituitary tumour at the time of PA diagnosis [5]. Among people with a pituitary adenoma, the incidence of PA varies from 1.5% to 27.7% [1,2,6,7,8]. 

The pathophysiology of PA is still debated. Several theories have been proposed considering the unique anatomy of the hypophyseal vascularization, but more probably PA is explained by a combination of multiple factors [9,10,11]. The enlarging pituitary tumour can either cause vascular compression or exceed blood supply, or its critical perfusion pressure can be below normal arterial pressure, leading to haemorrhage, infarction or ischemia [9,10,11]. This phenomenon causes the acute expansion of the pituitary tumour, with consequent occurrences of neurological signs [1,3,10]. In human medicine, there are few isolated reports of haemorrhagic lesions occurring in a normal pituitary gland, especially during pregnancy [9,10]. In these cases, PA pathogenesis is supposed to be different and is not clearly understood [9,10].

The typical clinical presentation of PA in humans is defined by the hyperacute onset of headache, with or without consciousness impairment, and rapidly progressive visual disturbances [2,3,4].

In human medicine, magnetic resonance imaging (MRI) represents the gold standard for the diagnosis of PA, as it allows to identify haemorrhages in 88–91% with accuracy in the identification of the early haemorrhagic stage [12,13]. At the same time, computed tomography (CT) has been frequently used in emergency settings, as it represents a quick access to an immediate imaging diagnosis [2,14]. The outcome for human patients with PA is overall good with both medical and surgical treatment [3,13,15,16,17,18]. In particular, visual deficits and ocular palsies improve very rapidly following surgery, but can also resolve with conservative treatment after a longer period of time [3,13,15,16,17,18].

In veterinary medicine, clinical signs, imaging findings and histopathological features have been described in a few dogs and one cat [19,20,21,22]. Altered mental status, visual deficits, ophthalmoplegia and epileptic seizures represent the most frequently reported neurological signs [19,20,21,22].

The computed tomography (CT) characteristics of PA have been described in detail in a case series of four dogs with histopathological confirmation [20]. High-field MRI characteristics have been defined in one dog and one cat, with imaging findings similar to those described in human medicine [2,3,4,21,22]. Among these reported cases, only one dog survived for several months after PA diagnosis [22]. The remaining cases were all euthanized at the time of PA diagnosis or after a few days because of rapid clinical deterioration [20,21]. However, a description of a more conspicuous number of cases and relative outcome is lacking.

Hence, the aim of this study is to give a description of the clinical and neurological signs, advanced imaging findings and outcome of a large number of dogs with suspected PA.

## 2. Materials and Methods

### 2.1. Inclusion and Exclusion Criteria

This is an observational retrospective study. The study population included client-owned dogs.

Previous informed written consent was obtained from all dog owners.

All dogs with a history of onset of neurological signs and a complete neurological examination, which underwent MRI and/or CT examination with findings suggestive of PA at our clinic from 1 January 2006 until 31 December 2021, were considered for inclusion in the study.

Inclusion criteria and information required were: (1) a complete history and clinical examination, (2) a complete clinicopathological evaluation (for further study; results are not reported in this paper), (3) a detailed neurological examination performed by a senior neurologist during the first 24 h after referral, and (4) CT scans and/or MRI study of the head compatible with suspected PA.

Data regarding time of resolution of clinical signs, outcome during hospitalization and long-term follow-up at further rechecks were collected when available, using both medical records and telephonic interviews.

Dogs were excluded if the inclusion criteria were not complete, if no clinical or neurological signs were recorded and/or if the CT/MRI scan revealed concomitant intracranial diseases (such as neoplasia other than the pituitary tumour or meningoencephalitis). Two cases considered for inclusion were already described in previous veterinary reports [20,22]. All the procedures performed complied with the European legislation “on the protection of animals used for scientific purposes” (Directive 2010/63/EU) and with the ethical requirements of the Italian law (Decreto Legislativo 4 March 2014, n. 26).

### 2.2. Clinical and Neurological Evaluation

Clinical information concerning signalment, history, onset of clinical signs and any previously diagnosis of a pituitary mass was recorded.

The onset of clinical signs before presentation was defined as per-acute (when clinical signs developed in less than 24 h), acute (1–7 days), or chronic (over weeks or months).

Neurological findings were reviewed. Furthermore, the presence of epileptic seizures was recorded. Data regarding hyperesthesia were carefully assessed with specific attention on the localisation, described as located at the level of head, cervical, thoracic or lumbar spine.

### 2.3. Diagnostic Imaging Findings

CT examinations were performed either using a 16-CT scanner (GE Medical Systems, Lightspeed 16, Milan, Italy) from 2006 to 2013 or a Dual Source CT scanner from 2014 to 2021 (128 × 2 DSCT, Somatom Definition Flash or 192 × 2 Somatom Force, Siemens, Erlangen, Germany).

MRI examinations were performed either using a 0.4 Tesla MRI (Aperto Lucent, SN, X418, Hitachi) from 2015 to 2018 or a 3 Tesla MRI (MAGNETOM Skyra, Siemens, Erlangen, Germany) from 2018 to 2021.

All MRI and CT images were retrieved from the Picture Archiving Communication System (PACS) and analysed using a dedicated freestanding workstation and vendor-specific postprocessing software (SyngoVia, Siemens, Erlangen, Germany).

CT images were examined looking for pre-contrast lesion heterogeneity and hyperattenuation, degree of contrast enhancement and presence of hypovascular areas. MRI images were evaluated for signal intensity and homogeneity (T1-weighted (T1W), T2-weighted (T2W) and fluid attenuated inversion recovery (FLAIR)). Signal intensities were given with respect to normal grey matter unless stated otherwise. Signal voids in Gradient Echo-T2 * (GE-T2 *) or Susceptibility Weighted Imaging (SWI) were recorded, as well as contrast enhancement homogeneity. Diffusion weighted imaging (DWI) and apparent diffusion coefficients (ADC) were performed using only high field MRI. DWI was acquired with a b value of 0 and 1000.

Mass effect was recorded for both CT and MRI studies. The pituitary-to-brain area ratio (P:B ratio) was calculated in transverse brain sections using preferentially CT images, but also MRI when brain CT was not available, as previously described [23,24]. Measured masses were then classified as enlarged or non-enlarged pituitary detectable masses according to P:B ratio value (>0.31 or <0.31, respectively), as previously stated [25].

Finally, concordance between the two procedures was reported considering conclusions made on final reports.

All CT scans and MRI studies were revised by the Head of the Diagnostic Imaging department (G.B.: a DVM, PhD, with a 20-year experience in advanced imaging), a DVM specialist in diagnostic imaging with a second level mater degree (G.D.S.), a neurology (ECVN) Resident (G.G.) and a board-certified neurologist (M.M.).

### 2.4. Outcome

Treatments were recorded including the ones used in the emergency setting.

The time of neurological signs’ resolution was recorded once available and classified as less than 24 h, between 24 and 48 h, between 48 and 72 h, and more than 72 h.

Long-term follow-up was recorded when available, using both medical records and phone calls to owners. Survival time was assessed, and then divided in less than one month, between one and six months, between six months and one year, and more than one year. The cause of death was recorded when available and classified as related to either PA or the presence of a pituitary mass, or due to an unrelated cause.

## 3. Results

### 3.1. Case Population

From the review of the medical records, 24 cases were initially included for data analysis and considered for inclusion. Two dogs were excluded because CT images did not clearly identify haemorrhage or infarction originating from a pituitary mass. In two other cases, imaging findings were indicative of haemorrhage within the third ventricle, but as there was no clear evidence of hypophyseal origin, and these two dogs were also excluded. One dog was excluded since no clinical or neurological signs were reported at the time of PA diagnosis. In this case, CT was performed to control an already known pituitary mass and did not identify signs of acute haemorrhage.

Nineteen dogs finally met the inclusion criteria. The breed distribution was mixed breed (9/19), Labrador Retrievers (3/19), English Bulldog (1/19), Beagle (1/19), Boxer (1/19), Springer Spaniel (1/19), Corso dog (1/19), Italian Hound (1/19) and Italian Bracco dog (1/19). Median body weight was 27 kg (range of 4.7–43 Kg). The study group comprised 11 females, of which 8 were spayed, and 8 males, of which 7 were intact.

The median age at the time of imaging diagnosis of PA was 10.7 years (range of 3.2–13.7 years). In 4 dogs (4/19), a pituitary mass was previously diagnosed. In these cases, the median age at the pituitary mass diagnosis was 9 years (range of 8.5–11 years).

### 3.2. Onset of Clinical Signs

The median age at onset of clinical signs was 10.6 years (range of 3.2–13.3 years).

The majority of dogs started showing clinical signs at the time of referral (11/19). A total of 14 dogs had per-acute onset of clinical signs (14/19). Of these, seven dogs displayed vestibular signs, seven dogs showed behavioural changes and four dogs exhibited epileptic seizures.

Three dogs were referred for acute onset of behavioural changes and obtundation (3/19), one of these also showed epileptic seizures. In the last 2 dogs, the onset of clinical signs was classified as chronic (2/19). Out of these, one dog was referred for the chronic onset of aggressive behaviour (Table 1). The remaining dog was referred for chronic onset of obtundation, weakness and loss of weight that had become worse over the previous 4 months and underwent CT for concomitant clinicopathologic findings suggestive of hyperadrenocorticism (1/5).

In addition, seven dogs showed accessory and non-specific signs (7/19).

Complete data concerning clinical signs’ onset are shown in Table 1.

### 3.3. Neurological Findings

Complete data regarding the neurological examination findings are shown in Table 2 and Appendix A.

The neurological examination revealed mainly behavioural and mental status alterations. Behavioural abnormalities were found in 11 dogs. All dogs with altered mental status showed obtundation (7/19).

Posture examination did not show any relevant alteration in the majority of dogs (18/19). Only one dog showed ventroflexion of the neck and concomitant neck pain (1/19). Gait analysis revealed alterations in 4 dogs (4/19).

Postural reaction deficits were described in 5/19 cases. Of those, 3/5 showed proprioceptive deficits on all four limbs.

Deficits at cranial nerves examination were found in 4 dogs (4/19) and concerned only the neuro-ophthalmological examination. In particular, menace response was bilaterally absent in 3 dogs (3/4) and reduced only on one side in 1 dog (1/4). Of dogs with bilaterally reduced menace response, one dog (1/4) showed internal ophthalmoplegia with bilateral mydriasis and bilaterally absent direct and consensual pupillary light reflex (PLR). Another dog (1/4) showed internal ophthalmoparesis with bilaterally reduced direct and consensual PLR. No deficits at the examination of other cranial nerves were reported.

Epileptic seizures were described in 6 dogs (6/19). Specifically, 5 dogs showed epileptic seizures at the time of referral, while one dog started having epileptic seizures during hospitalization three days after PA diagnosis (1/6).

Pain and/or hyperalgesia were evident in 3 dogs (3/19). In particular, pain on manipulation of the neck was elicited in 2 dogs (2/3).

Three dogs did not show any abnormal findings at the time of neurological examination (3/19). Two of them were referred for per-acute onset of episodes of vestibular signs, which were resolved in less than 24 h and between 48 and 72 h, respectively. One dog was referred for the per-acute onset of epileptic seizures (1/3).

### 3.4. Imaging Findings

Eighteen out of nineteen dogs underwent CT examination. Nine dogs underwent MRI. Eight dogs underwent both procedures. In all cases, a pituitary mass was identified.

Complete data concerning CT and MRI characteristics are shown in Table 3 and Table 4, respectively.

In pre-contrast TC scans, all lesions appeared hyperattenuating and heterogeneous. Twelve (12/18) showed hyperattenuating focal areas (Figure 1 and Figure 2). Heterogeneous contrast enhancement was evident in all 18 CT studies and hypovascular areas were identified in 15 cases after contrast medium administration (15/18).

Concerning the MRI studies, the majority of lesions appeared heterogeneously hyperintense in T2W and FLAIR images (6/9). T1W signal appeared heterogeneously hypointense in the majority of cases (4/9). In four cases, T1W signal was isointense (4/9) either heterogeneously (2/4) or homogeneously (2/4). Contrast enhancement was heterogeneous in all cases.

DWI and ADC maps were performed in 6 cases (6/9), using only high field MRI. In 3 studies, DWI showed hypointense foci (3/6). Corresponding diffusion coefficients in ADC maps were increased in two cases and decreased in one. In two cases, DWI highlighted hyperintense foci with a decreased diffusion coefficient in ADC map (2/6) (Figure 3). One case presented with isointense DWI and ADC map (1/6).

GE-T2 * or SWI were performed in seven cases. Signal voids were identified in 5 cases (5/7); hence, they were considered highly suggestive of haemorrhage. In four MRI studies, the evaluation of standard sequences (T1W, T2W and FLAIR) allowed the identification of pituitary lesions characterized by a heterogeneous aspect with signal referrable to haemorrhage or ischaemic necrosis. One dog with negative GE-T2 * did not show any other MRI signal suggestive of haemorrhage. However, in this case, PA was clearly identified using CT. In particular, in pre-contrast CT images, the lesion appeared heterogeneous and with hyperattenuating foci highly suggestive of intralesional haemorrhage.

Signs of mass effect were found in 11 out of 19 cases using both CT and MRI. The median pituitary mass height was 12.8 mm (range of 5.2–27 mm) and median P:B ratio was 0.78 (range of 0.33–1.35). All cases were then classified as enlarged pituitary detectable masses.

Considering dogs that underwent both CT and MRI, the imaging findings were similarly described in seven out of eight studies.

### 3.5. Treatment and Outcome

Nine dogs received some medical treatment (9/19). An emergency treatment was started with prednisolone in six dogs (6/9) and/or antiepileptic medications (2/9), while one dog was administered mannitol in order to reduce the intracranial pressure (1/9). Two dogs were treated with trilostane for concomitant hyperadrenocorticism (2/9). One dog underwent radiotherapy in addition to the medical therapy (1/9).

Information concerning neurological signs resolution were available in 15 cases (15/19). Neurological signs resolved in less than 24 h in seven dogs (7/15). Among these, four dogs showed vestibular signs (4/7) and four dogs showed epileptic seizures (3/7), which were associated with internal ophthalmoplegia in one dog. The majority of dogs (6/7) with resolution of the neurological signs within 24 h did not receive any medical treatment, while one dog (1/7) received prednisolone. One of these dogs was unfortunately lost for long-term follow-up.

Two dogs, showing behavioural abnormalities and epileptic seizures, respectively, had neurological sign resolution within 48 h (2/15). In two dogs, neurological signs were resolved in 72 h (2/15). In particular, they showed, respectively, vestibular episodes (1/2) and obtundation associated with behavioural changes and cervical pain (1/2). One dog with behavioural changes, reduced menace response on one side and diffuse hyperalgesia on palpation of the entire spine did not recover completely from the neurological signs and behavioural changes persisted (1/15).

Three dogs died during hospitalisation after PA diagnosis (3/15). Specifically, two dogs were referred for epileptic seizures and did not respond to the antiepileptic treatment. One of them was euthanized the day after the PA diagnosis, while the other one died from cardiovascular arrest 3 days after PA diagnosis. The remaining dog was euthanised 4 days after PA diagnosis because of the rapid worsening of signs of increased intracranial pressure with no response to medical treatment. Overall, twelve dogs survived at the time of diagnosis of PA (12/15).

Information regarding long-term follow-up were available for 11 dogs (11/19). Five dogs were still alive at the time of data collection (5/11). The median long-term follow-up time was of 18 months (range of 1–102 months). The lower long-term follow-up time was of 1 month. Four dogs were euthanised because of the progressive worsening of the clinical signs related to the growing pituitary mass (4/11). In particular, one dog survived 1 month after PA diagnosis (1/4), one dog survived between 6 months and 1 year (1/4), and two dogs survived more than 1 year (2/4). Two dogs were euthanised, respectively, 7 months and 2 years after PA diagnosis for totally unrelated causes (2/11). The overall median survival time was 7 months (range of 0–102 months) from the time of PA diagnosis. Considering only dogs that survived at the time of diagnosis and were discharged from hospitalisation, the median survival time was 15.5 months (range of 1–102 months). One dog underwent radiotherapy treatment with a survival time of 8.5 years.

Detailed information concerning the resolution time of neurological signs and long-term follow-up can be found in Table 5.

## 4. Discussion

PA is diagnosed by combining the acute onset of neurological signs with typical diagnostic imaging findings suggestive of haemorrhage or infarction in a normal or tumoral pituitary gland [1,2,3,4]. In human medicine, 95–100% of patients with PA show sudden onset of headaches as the most common clinical sign, typically occurring in the retrobulbar or frontal region, followed by visual deficits, ocular palsy, nausea and vomiting [2,12,15,26,27]. In the veterinary literature, a total of six dogs and one cat with histopathological and/or imaging diagnosis of PA are described, with behavioural changes, mental status obtundation, visual deficits and epileptic seizures being the most common neurological findings reported [19,20,21,22].

In the present study, the onset of clinical signs was per-acute in the vast majority of cases and was frequently characterized by neurological signs. In particular, behavioural changes, vestibular signs and epileptic seizures were clinical and neurological signs mostly represented at the time of clinical onset.

Behavioural change was the most common clinical sign found in this study, as previously reported [20,22]. Behavioural change is a common finding in dogs with a pituitary detectable mass and is associated with both the size of the pituitary mass and the presence of mass effect. In the case of PA, haemorrhage can lead to a sudden increase in pression on the surrounding brain structures causing secondary mass effect [25]. Furthermore, the presence of blood in the subarachnoid space and/or meningeal stretching causes meningeal irritation with consequent headache and neck stiffness in human patients [2,28]. In veterinary medicine, the assessment of headaches is very challenging and subjective. It is usually suspected in conjunction to neck pain or pain on palpation of the base of the head, secondary to compression or stretching of meninges or of cerebral vasculature [29]. In this case series, cervical pain was noted in only two patients, hence it cannot be considered a common presenting sign. However, obtundation was frequently described and, from this perspective, it can represent a clinical manifestation of headaches.

According to the literature, neuro-ophthalmologic deficits have been described quite frequently in patients with PA [20,21,22]. In the present study, cranial nerve examination revealed only menace response and PLR deficits. The evaluation of remaining cranial nerves did not show any abnormalities.

However, only two dogs showed specifically internal ophthalmoparesis or ophthalmoplegia. In dogs, the hypophyseal peduncle is loosely encircled by a thin layer of dura mater, thus favouring dorsal tumour growth toward the diencephalon, and preventing the compression of the optic chiasm rostrally [30,31]. As a result, a visual impairment in dogs should be less frequent than in human patients and an involvement of oculomotor parasympathetic fibres, resulting in internal ophthalmoparesis/plegia, seems to be more likely since they run medially over the pituitary fossa. The occurrence of neuro-ophthalmologic deficits is not common in our study compared to the veterinary literature. This difference can result from the size of the original pituitary mass, which experienced haemorrhage, or may be related to the haemorrhage extent.

In human medicine, CT is essential in emergency settings in order to obtain a quick diagnosis of pituitary tumour with associated haemorrhage and exclude other differentials [2,14]. However, MRI constitutes the diagnostic imaging modality of choice for PA diagnosis since it is able to identify a pituitary tumour in 100% of cases and PA in 88% of patients compared to CT (93% and 21% of cases, respectively) [3,4,12]. The principal CT and MRI characteristics of PA were described in detail in a large number of patients for the first time in this study. CT images in dogs with PA have already been described, even if only once in conjunction with MRI [20,22].

The diagnosis of vascular lesion using either CT or MRI is not always straightforward. CT is not always able to differentiate subacute or chronic haemorrhagic foci from necrotic foci within a tumour [20,32]. In MRI, an evaluation of standard sequences in conjunction with specific sequences for the identification of vascular events is fundamental in order to properly describe a haemorrhagic or ischemic lesion. In particular, the evaluation of T2 * and SWI sequences allows the prompt identification of haemorrhagic foci. Instead, DWI sequences and respective ADC maps provide the recognition of ischaemic lesions. The increasing use of this advanced sequences will help a more accurate diagnosis and aid in faster decision making for PA treatment in the future. In human medicine, these sequences proved to be fundamental in order to obtain a correct diagnosis and define the most appropriate treatment for the patient, for example, choosing surgical treatment [32]. Unfortunately, in veterinary medicine, the most appropriate therapeutical approach has not been defined at the present moment. However, making a more accurate diagnosis will help the decision-making process of the clinician.

To the authors’ knowledge, there seems to be no correlation between tumour size and occurrence of PA in the current literature. However, neurological signs have been previously associated with the evidence of mass effect [33] and to tumour size [34,35]. In our study, the diagnostic imaging findings revealed the presence of mass effect in the majority of cases and a P:B ratio > 0.31 in all dogs. Hence, it can be speculated that this evidence may either represent a casual finding or that enlarged pituitary detectable masses may be more likely to bleeding. A potential association among tumour size and onset of clinical signs, neurological presentation or time of resolution of clinical signs could was investigated. Unfortunately, since all included cases had large pituitary masses (P:B ratio > 0.31), such investigation was not possible.

Finally, this case series highlights the high agreement between CT and MRI in identifying PA. From this perspective, the use of CT can be strongly suggested in an emergency setting, in the case of the patient not being stable enough to go through an MRI study, or when MRI is not available.

In our study, CT seemed to be more sensible in identifying the haemorrhage in one case. Both CT and MRI were performed within 24 h from the onset of clinical signs. CT examination showed a pituitary lesion that appeared inhomogeneously hyperattenuated, indicating haemorrhagic foci. Contrast enhancement was inhomogeneous and revealed hypovascular areas suggestive of ischeamic necrosis. At the MRI examination, the identified lesion was heterogeneously isointense to hypointense in T1W, heterogeneously hyperintense in T2W and FLAIR sequences, and contrast enhancement was heterogeneous, but a clear signal void was not identified on the GE-T2* sequence. However, the MRI aspect of acute haemorrhages is usually characterized by T1W hypo/isointensity, T2W hypointensity, DWI and ADC map hypointensity and marked signal void in GE-T2 * sequences [36]. Hence, in this specific case, DWI and ADC map may have been useful in order to confirm the haemorrhagic lesion. These sequences revealed to be fundamental in describing early stages of haemorrhage in human cases of PA [33,37]. Unfortunately, in our case, DWI and ADC map were not performed since that study was performed with a low field MRI (0.4 T). Furthermore, in the hyperacute and acute phase, haemorrhage appears as a rim of hypointense signal around an isodense core on GE-T2* sequence [36]. In this case, a possible explanation for the negativity of the GE-T2* sequence could be an early stage of haemorrhage associated with the relatively reduced dimensions of the pituitary lesion and the low field MRI.

The outcome of patients with PA is basically unknown in veterinary medicine, since the majority of cases described were euthanised at the time of diagnosis. In this study, the outcome of dogs with PA is described for the first time. The vast majority of dogs recovered from per-acute and acute signs and being discharged from hospitalization. This is similar to human medicine, where the outcome is generally favourable. Human patients with PA are treated conservatively with corticosteroids and supportive therapies in an emergency setting, but a surgical approach with the resection of the pituitary mass is often taken into consideration, especially in the case of altered mental status, visual deficits and ophtalmoplegia [2,3,4,9]. However, patients can be safely managed conservatively if no or stable visual deficits are present [2,13,17,18]. Pituitary function outcome and deficiencies have been reported to be almost the same with both surgical and conservative treatment [13,15,16,17,18]. Surgical transsphenoidal hypophysectomy is not routinely performed in dogs with PA, especially in an emergency setting, since trained medical staff is needed both for the surgical procedure and for an adequate intensive care treatment postoperatively. Transsphenoidal hypophysectomy has been shown to be helpful and safe for the treatment of both pituitary hyperadrenocorticism and endocrinologically inactive pituitary macrotumours [38,39]. Hence, in the future, it may be considered a tool for the treatment of PA as well. However, the results of the present study seem to ensure that a conservative treatment can be safely used in patients with a diagnosis of PA. The patient should be given some time to recover from the vascular event, considering euthanasia in cases where there is no recovery from severe neurological signs.

This study has some limitations, principally due to its retrospective nature. The PA images were acquired with different imaging modalities, which may affect images interpretation. However, imaging studies were always evaluated by imaging specialists. One other limitation is due to the case selection, as we only included dogs that retrospectively had a PA diagnosis made on a combination of diagnostic imaging findings and a complete neurological examination. Thus, this could have led to a bias in the selection of the study population, excluding asymptomatic dogs. However, this decision was made with the precise purpose to describe the neurological presentation of dogs with PA. Medical records regarding the outcome and long-term follow up were only available in a limited number of cases, limiting a reliable outcome evaluation. This difference in data availability was due to the inability to contact some owners, probably due to the long-time intercourse to the case presentation. Finally, as the overall outcome was favourable, a definitive histopathologic diagnosis is missing. A definitive histopathologic diagnosis would have certainly allowed a better discrimination identifying pituitary masses and distinguishing them from other differentials. In particular, pituitary masses should be distinguished from pathologies with either an inflammatory or vascular nature, or other tumours, such as meningiomas or metastatic lesions. However, these differentials usually have specific shape, location and diagnostic imaging features. Furthermore, in this study, many patients already had an imaging diagnosis of a pituitary mass before presentation and were selected also on the basis of diagnostic imaging findings compatible with PA.

## 5. Conclusions

The present study represents a case series of dogs with a presumed diagnosis of PA based on imaging findings. The outcome of dogs with PA is favourable in the majority of our cases. Almost half of the dogs recovered from clinical signs in less than 24 h. Hence, PA could be an underdiagnosed pathology in dogs and cats. From this perspective, imaging findings are fundamental to reach an early diagnosis in dogs with a sudden onset of common neurological abnormalities, such as behavioural changes, obtundation, vestibular signs, neuro-ophthalmological alterations and epileptic seizures.

## Figures and Tables

**Figure 1 vetsci-09-00191-f001:**
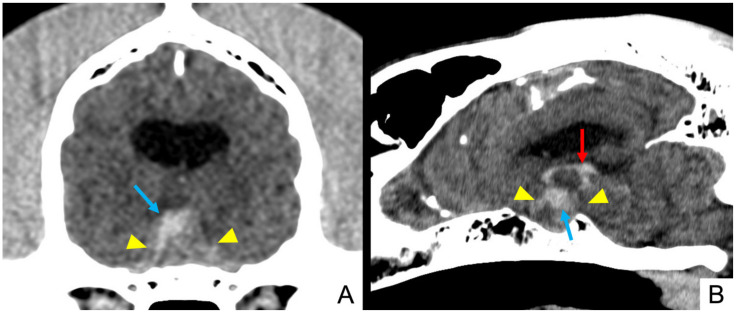
Dog 4: pre-contrast multiplanar reconstruction of brain CT scan showing an enlarged pituitary mass (yellow arrowheads) characterized by a wide intralesional hyperattenuating area (light blue arrows), suggesting intralesional haemorrhage extending in the third ventricle (red arrow). (**A**) Transverse plane and (**B**) sagittal plane.

**Figure 2 vetsci-09-00191-f002:**
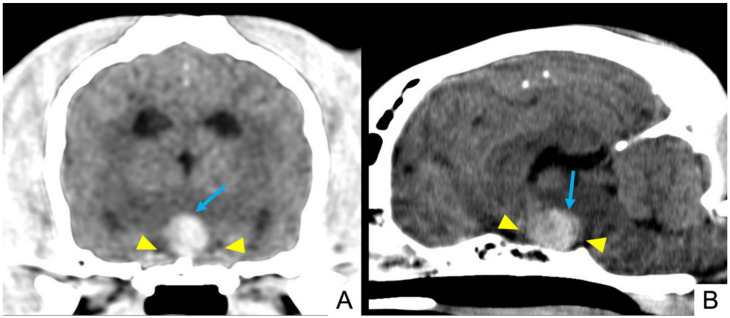
Dog 3: pre-contrast multiplanar reconstruction of brain CT scan, showing an enlarged pituitary mass (yellow arrowheads) characterized by heterogeneous aspect with hyperattenuating areas (light blue arrows), suggesting intralesional haemorrhage. (**A**) Transverse plane and (**B**) sagittal plane.

**Figure 3 vetsci-09-00191-f003:**
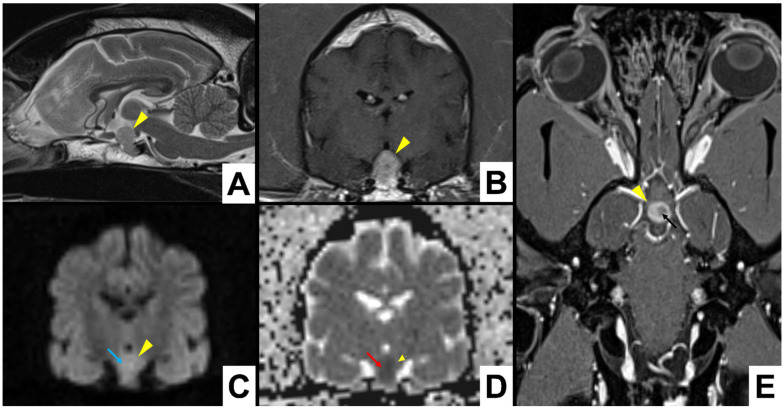
Dog 1: A total of 3 Tesla MRI scan of a pituitary mass (yellow arrowheads), showing hyperintense signal in DWI (light blue arrow) and hypointense in the corresponding ADC map (red arrow), suggesting acute pituitary haemorrhage, heterogeneous contrast enhancement with an hypointense area (black arrow). (**A**) T2W in sagittal plane as reference position; (**B**) T1W post-contrast; (**C**) DWI b1000; (**D**) ADC map transverse sections; (**E**) T1W dorsal section.

**Table 1 vetsci-09-00191-t001:** Description of signalment; onset classification of clinical signs (per-acute/acute/chronic); owner’s main complaint; epileptic seizures at clinical sign’s onset; evidence of accessory clinical signs.

Signalment	Onset (Per-Acute/Acute/Chronic)	Main Complain	Epileptic Seizures	Accessory Signs
1. Mixed breed MN, 9.75 y, 24.6 Kg	Per-acute	Balance loss	No	\
2. Italian hound F, 11.5 y, 14.8 Kg	Per-acute	Balance loss	No	Pica
3. Mixed breed FN, 12.7 y, 4.7 Kg	Per-acute	Balance loss	No	\
4. Italian Bracco dog F, 12.8 y, 27 Kg	Per-acute	Compulsion, circling, obtundation	No	\
5. Mixed breed FN, 12.7 y, 30 Kg	Per-acute	Vocalization, pathologic nystagmus	No	\
6. Beagle FN, 7.6 y, 16.6 Kg	Per-acute	Balance loss, pathologic nystagmus	No	\
7. Labrador Retriever F, 5 y, 31.2 Kg	Per-acute	Anxious behaviour	Yes	PU/PD, weakness
8. Labrador Retriever FN, 10.8 y, 35 Kg	Per-acute	\	Yes	\
9. Mixed breed FN, 5.8 y, 21 Kg	Chronic	Aggressive behaviour	No	\
10. Labrador Retriever FN, 8.6 y, 29 Kg	Per-acute	\	Yes	\
11. Boxer FN, 11.4 y, 27.4 Kg	Per-acute	Vocalizations, circling	No	\
12. Springer Spaniel M, 13.7 y, 18.7 Kg	Acute	Disorientation, anxious behaviour, obtundation	Yes	\
13. Mixed breed M, 11.1 y, 33 Kg	Per-acute	Circling	No	Weight loss
14. Mixed breed M, 9.8 y, 14.5 Kg	Per-acute	Disorientation, obtundation, balance loss	No	Tremors
15. Corso dog FN, 9.4 y, 43 Kg	Per-acute	Balance loss	No	\
16. English Bulldog M, 6.1 y, 28 Kg	Per-acute	Aggressive behaviour, vocalization, obtundation	No	Dysorexia, tremors
17. Mixed breed M, 9.3 y, 9.3 Kg	Chronic	Obtundation	No	Weakness, weight loss, tremors
18. Mixed breed M, 3.2 y, 15 Kg	Per-acute	Non-specific changes from normal behaviour	Yes	\
19. Mixed breed M, 11.1 y, 27.6 Kg	Acute	Vocalization, obtundation	No	PU/PD

**Table 2 vetsci-09-00191-t002:** Detailed neurological evaluation of the study population.

Neurologic Examination	Behaviour	Mental Status	Posture	Gait	Postural Reactions	Cranial Nerves	Epileptic Seizures	Hyperalgesia
Results	Normal (*n* = 8) Altered (*n* = 11)	Normal (*n* = 12) Altered (*n* = 7)	Normal (*n* = 18) Altered (*n* =1)	Normal (*n* = 15) Altered (*n* = 4)	Normal (*n* = 14) Altered (*n* = 5)	Normal (*n* = 15) Altered (*n* =4)	No (*n* = 13) Yes (*n* = 6)	No (*n* = 16) Yes (*n* = 3)
Detail	Disorientation (*n* = 6) Compulsion (*n* = 4) Circling (*n* = 3) Aggressive behaviour (*n* = 3) Head pressing (*n* = 2) Anxiety (*n* = 2) Vocalizations (*n* = 2)	Obtundation (*n* = 7)	Neck ventroflexion (*n* = 1)	Hindlimbs hypometria (*n* = 1) Hindlimbs proprioceptive ataxia (*n* = 1) Proprioceptive ataxia 4 limbs (*n* = 1) Tetraparesis (*n* = 1)	4 limbs (*n* = 3) 1 posterior limb (*n* = 1) Left side (*n* = 1)	Altered menace response (*n* = 4) Internal ophtalmoparesis/plegia (*n* = 2)		Cervical (*n* = 2) Diffuse (*n* = 1)

**Table 3 vetsci-09-00191-t003:** Description of CT characteristics. In case #2, a CT was not performed.

Signalment	Pre-Contrast Heterogeneity	Pre-Contrast Hyperattenuation	Hyperattenuating Foci	Contrast Enhancement	Post-Contrast Hypovascular Areas	CT Diagnosis
1. Mixed breed MN, 9.75 y, 24.6 Kg	Yes	Yes	Yes	Heterogeneous	Yes	Pituitary mass with intralesional haemorrhage
3. Mixed breed FN, 12.7 y, 4.7 Kg	Yes	Yes	Yes	Heterogeneous	Yes	Pituitary mass with intralesional haemorrhage
4. Italian Bracco dog F, 12.8 y, 27 Kg	Yes	Yes	Yes	Heterogeneous	Yes	Pituitary mass with intralesional haemorrhage
5. Mixed breed FN, 12.7 y, 30 Kg	Yes	Yes	Yes	Heterogeneous	Yes	Pituitary mass with intralesional haemorrhage
6. Beagle FN, 7.6 y, 16.6 Kg	Yes	Yes	Yes	Heterogeneous	Yes	Pituitary mass with intralesional haemorrhage
7. Labrador Retriever F, 5 y, 31.2 Kg	Yes	Yes	Yes	Heterogeneous	No	Pituitary mass with intralesional haemorrhage
8. Labrador Retriever FN, 10.8 y, 35 Kg	Yes	Yes	Yes	Heterogeneous	Yes	Pituitary mass with intralesional haemorrhage
9. Mixed breed FN, 5.8 y, 21 Kg	Yes	Yes	Yes	Heterogeneous	Yes	Pituitary mass with intralesional haemorrhage
10. Labrador Retriever FN, 8.6 y, 29 Kg	Yes	Yes	Yes	Heterogeneous	Yes	Pituitary mass with intralesional haemorrhage
11. Boxer FN, 11.4 y, 27.4 Kg	Yes	Yes	No	Heterogeneous	No	Pituitary mass with intralesional haemorrhage
12. Springer Spaniel M, 13.7 y, 18.7 Kg	Yes	Yes	No	Heterogeneous	Yes	Pituitary mass with intralesional haemorrhage
13. Mixed breed M, 11.1 y, 33 Kg	Yes	Yes	No	Heterogeneous	Yes	Pituitary mass with intralesional haemorrhage
14. Mixed breed M, 9.8 y, 14.5 Kg	Yes	Yes	No	Heterogeneous	Yes	Pituitary mass with intralesional haemorrhage
15. Corso dog FN, 9.4 y, 43 Kg	Yes	Yes	Yes	Heterogeneous	No	Pituitary mass with intralesional haemorrhage
16. English Bulldog M, 6.1 y, 28 Kg	Yes	Yes	No	Heterogeneous	Yes	Pituitary mass with intralesional haemorrhage and necrosis
17. Mixed breed M, 9.3 y, 9.3 Kg	Yes	Yes	Yes	Heterogeneous	Yes	Pituitary mass with intralesional haemorrhage and necrosis
18. Mixed breed M, 3.2 y, 15 Kg	Yes	Yes	Yes	Heterogeneous	Yes	Pituitary mass with intralesional haemorrhage and necrosis
19. Mixed breed M, 11.1 y, 27.6 Kg	Yes	Yes	No	Heterogeneous	Yes	Pituitary mass with intralesional haemorrhage

**Table 4 vetsci-09-00191-t004:** Description of the MRI signal characteristics of the identified lesions. All signals are presented with respect to the normal grey matter. FLAIR, fluid attenuated inversion recovery; T2 */SWI, susceptibility weighted imaging, indicating presence (yes) or absence (no) of signal voids; DWI, diffusion weighted imaging; ADC, apparent diffusion coefficient. The “\” symbol indicates that the sequence was not performed. In cases #3,4,10,13,14,15,16,17,18,19, MRI was not performed.

Signalment	T1-Weighted	T2-Weighted	FLAIR	T2*/SWI	DWI	ADC	Contrast Enhancement	MRI Diagnosis
1. Mixed breed MN, 9.75 y, 24.6 Kg	Homogeneous isointense	Heterogeneous isointense	Heterogeneous isointense	No	Hyperintense	Hypointense	Heterogeneous	Pituitary mass with different stages of intralesional haemorrhage
2. Italian hound F, 11.5 y, 14.8 Kg	Heterogeneous isointense	Heterogeneous isointense	Heterogeneous isointense	Yes	Hyperintense	Hypointense	Heterogeneous	Pituitary mass with intralesional haemorrhage
5. Mixed breed FN, 12.7 y, 30 Kg	Homogeneous isointense	Heterogeneous isointense	Heterogeneous isointense	Yes	Isointense	Isointense	Heterogeneous	Pituitary mass with intralesional haemorrhage
6. Beagle FN, 7.6 y, 16.6 Kg	Heterogeneous hypointense	Heterogeneous hyperintense	Heterogeneous hyperintense	Yes	Hypointense	Hypointense	Heterogeneous	Pituitary mass with intralesional haemorrhage
7. Labrador Retriever F, 5 y, 31.2 Kg	Heterogeneous hypointense	Heterogeneous hyperintense	Heterogeneous hyperintense	\	Hypointense	Hyperintense	Heterogeneous	Pituitary mass with intralesional haemorrhage
8. Labrador Retriever FN, 10.8 y, 35 Kg	Heterogeneous hypointense	Heterogeneous hyperintense	Heterogeneous hyperintense	No	\	\	Heterogeneous	Pituitary mass
9. Mixed breed FN, 5.8 y, 21 Kg	Heterogeneous hypointense	Heterogeneous hyperintense	Heterogeneous hyperintense	Yes	\	\	Heterogeneous	Pituitary mass with intralesional haemorrhage
11. Boxer FN, 11.4 y, 27.4 Kg	Homogeneous hyperintense	Heterogeneous hyperintense	Heterogeneous hyperintense	\	\	\	Heterogeneous	Pituitary mass with intralesional haemorrhage
12. Springer Spaniel M, 13.7 y, 18.7 Kg	Heterogeneous isointense	Heterogeneous hyperintense	Heterogeneous hyperintense	Yes	Hypointense	Hyperintense	Heterogeneous	Pituitary mass with intralesional haemorrhage

**Table 5 vetsci-09-00191-t005:** Description of signalment; onset classification of clinical signs (per-acute/acute/chronic); time needed to recover from neurological signs when available; long-term follow-up specifying cause of death (related/unrelated to PA or presence of a pituitary mass), if the patient died during hospitalization (D) and survival time in months (m) when available; performed MRI and/or CT; P:B ratio value; presence of mass effect (yes/no).

Signalment	Onset (Per-Acute/Acute/Chronic)	Recovery Time	Long Term Follow-Up	MRI/CT	P:B Ratio	Mass Effect (Yes/No)
1. Mixed breed MN, 9.75 y, 24.6 Kg	Per-acute	<24 h	Alive (11 m)	MRI CT	0.52	No
2. Italian hound F, 11.5 y, 14.8 Kg	Per-acute	<24 h	Alive (6 m)	MRI	0.35	No
3. Mixed breed FN, 12.7 y, 4.7 Kg	Per-acute	<24 h	Alive (18 m)	CT	0.75	Yes
4. Italian Bracco dog F, 12.8 y, 27 Kg	Per-acute	48–72 h	Unrelated survival time: death 7 m	CT	0.78	Yes
5. Mixed breed FN, 12.7 y, 30 Kg	Per-acute	<24 h	Alive (32 m)	MRI CT	0.37	No
6. Beagle FN, 7.6 y, 16.6 Kg	Per-acute	48–72 h	Related survival time: death 23 m	MRI CT	0.48	No
7. Labrador Retriever F, 5 y, 31.2 Kg	Per-acute	<24 h	Alive (34 m)	MRI CT	0.35	No
8. Labrador Retriever FN, 10.8 y, 35 Kg	Per-acute	D	Related death for cardiovascular arrest 3 days after PA diagnosis	MRI CT	0.39	No
9. Mixed breed FN, 5.8 y, 21 Kg	Chronic	Persistent behavioural changes	Related survival time: death 8 m	MRI CT	1.3	Yes
10. Labrador Retriever FN, 8.6 y, 29 Kg	Per-acute	<24 h	Unknown	CT	0.33	No
11. Boxer FN, 11.4 y, 27.4 Kg	Per-acute	24–48 h	Related survival time: death 1 m	MRI CT	1.2	yes
12. Springer Spaniel M, 13.7 y, 18.7 Kg	Acute	<24 h	Unrelated survival time: death 25 m	MRI CT	0.8	Yes
13. Mixed breed M, 11.1 y, 33 Kg	Per-acute	D	Euthanised 4 days after PA diagnosis	CT	0.88	Yes
14. Mixed breed M, 9.8 y, 14.5 Kg	Per-acute	Unknown	Unknown	CT	1.1	Yes
15. Corso dog FN, 9.4 y, 43 Kg	Per-acute	Unknown	Unknown	CT	0.44	No
16. English Bulldog M, 6.1 y, 28 Kg	Acute	Unknown	Unknown	CT	1.35	Yes
17. Mixed breed M, 9.3 y, 9.3 Kg	Chronic	24–48 h	Related survival time: death 102 m	CT	1.3	Yes
18. Mixed breed M, 3.2 y, 15 Kg	Per-acute	D	Euthanised 1 day after PA diagnosis	CT	1.3	Yes
19. Mixed breed M, 11.1 y, 27.6 Kg	Acute	Unknown	Unknown	CT	1.03	Yes

## Data Availability

The data presented in this study are available on request from the corresponding author. The data are not publicly available due to privacy.

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
