# Peer review of "Suspected Pituitary Apoplexy: Clinical Presentation, Diagnostic Imaging Findings and Outcome in 19 Dogs"

_vetsci, 2022, doi:10.3390/vetsci9040191_

Round 1

Reviewer 1 Report

A brief summary 

The aim of the article was to describe the clinical and neurological signs, DI findings and outcome in dogs with imaging diagnosis of PA in a considerable number of dogs. Interresting and useful information is described. In addition, the number of cases for a case series is quite high. Neurology clinicians can find the information from this article useful for reading/interpreting the MRI scans and making the clinical decisions/discussing the prognosis with the pet owners. 

General concept comments

Article

A retrospective study with lacking homogeneous data in order to be able to draw strong consslusions. In addition, the histopathological confirmation is lacking (for very obvious and plausible reasons) and it should be stated that the diagnosis is a suspected diagnosis based on DI findings (or a radiological diagnosis). In addition, it is impossible to say for sure that all neurological signs, especially the transient ones, are attributable to the PA without performing a histopathological examination.

A clear description of the diagnostic imaging findings is missing and the reader might be confused how to identify the radiological findings in cases of PA. Associations between the affected pituitary gland findings (hemorrhages, necrosis, ischaemic infacrtions)as well as size of the mass and the history, clinical, neurological signs, and the outcome is missing. If any associations are identified, they will help clinicians to make clinical decision and communicate prognosis with the owners of affected animals. 

There is no hypothesis in the article. 

Specific comments

Abstract

Line 19 - please change 'good' outcome to 'favourable' in the entire text

Introduction

Line 61: Please provide a short overview of outcomes described so far in dogs suffering from PA in the introduction. It is important for the readers to know that in previously described cases some dogs died during the imaging process or their neurological status deteriorated and therefore they were euthanised. 

M&M

Line 71: what do you mean by ‘collection procedures’? Please specify.

Lines 81-82: only dogs with radiologically confirmed PA were included in the study? Please state it in the text.

Line 82: please state what was the minimum follow-up time period required to be included in  the study. Were the dogs only with long-term follow-up included?

Lines 83-84: if phone interview based outcome data collection was unavailable/unsucessful, were these patients still included in the study, please specify.

Lines 99-103: Please remove the detailed description of neurological examination in the M&M section. The description of neurological examination is provided in separate dedicated textbooks.

Lines 129-130: Please describe who were reviewing the CT and MR images - were they in all instances evaluated and reported by a board certified diagnostic imaging specialists?

Lines 116-121: Please describe in more details the parameters used to generate the MRI scans and please include the description of DWI parameters, especially the b values chosen for the DWI.

Line 129: do you mean between the two imaging modalities? Please specify.

Lines 137-138: dogs that died during hospitalisation for the causes related to PA should be included in the survival time assessment.

Results

Lines 148-149: why do you state that dogs were excluded because no clear hemorrhage was identified in the CT scans when PA is an umbrella term not only for hemorrhages? The sentence needs more specifications.

Lines 151-152: I missed a clear statement in the M&M section inclusion criteria that only dogs with radiological diagnosis of PA in conjunction of abnormal neurological examination/history of recent neurological signs were included. Please specify in the M&M section.

Line 155: change to ‘remaining’

Table 1: please specify in the table section long-term follow-up what was the time period of follow-up in each case (e.g., 7 months etc.)

Line 210-211: change ‘proprioceptive’ deficits to ‘postural reaction’ deficits in the entire text

Line 249: change ‘sequences’ to ‘images’

Lines 249-251: the sentence describing signal intensities in MR images needs to be rephrased

Line 263: please specify what was exactly found in the CT images in this case.

Lines 265-266: The P:B ratios are quite variable in case series population. Was the size of the affected pituitary gland/mass associated with the onset, severity and rapidity of resolution of the clinical signs? It might help to create groups of the sizes and investigate these associations

Line 302 - authors state that in 15 dogs the outcome was available. Please specify in results section why follow-up was not available in the rest 4 dogs included in the case series

 Tables 3 and 4: I do not find this type of data presentation very helpful to the reader. More useful tables would be describing imaging sequence findings in conjunction with each other and in addition, comparison to CT findings in the same cases with inclusion of actual representative images.

Please create a table with groups of MRI and CT patterns of changes detected in the pituitary mass

E.g.,

Group 1: 4 cases, suspected acute hemorrhage: T2W SI hypo, T1W SI iso, T2* signal void etc. (in all 4 cases the imaging findings are roughly the same) and pair them with CT findings (where available) as well.

Group 2: 5 cases, suspected acute hemorrhage and infarction: T2WI SI heterogeneous hypo and hyper etc.

Group 3: 2 cases, suspected haemorrhage and necrosis: etc.

Provide typical paired MRI and CT images for each (at least with bigger sample size) group as an example.,

Please investigate if if radiological finding patterns were associated with any clinical data (the rapidity, severity of neurological signs and/or rapidity of resolution of neurological signs and overall outcome)

 Line 324: what was the median and/or mean and range of follow-up time in dogs with available long term follow-up?

Discussion

Lines 346-349: sentences need to be rephrased

Lines 349-353: please divide the long sentence into two sentences with clear meaning. Use term ‘mass effect’ where appropriate

Line 356 add headache in animals is a subjective finding in many instances, please clarify in the text

Lines 362 and 364 ‘neuro-ophthalmic’ or ‘neuro-ophthalmological’? please be consistent

Line 372 please add ‘possible’ or ‘suspected’ vascular nature as there are no histopathological evidence available

Lines 381-382 please rephrase the sentence

Lines 390-392: please explain/discuss how the diagnoses of different types of PA made by use of advanced imaging in humans aid in guiding the patient treatment. How does it or does it not translate in canine patients?

Line 397: did the size of the pituitary tumour were associated with the rapidity, severity of neurological signs and/or rapidity of resolution of neurological signs and overall outcome? Please discuss and add these findings into result section.

Line 455: please add low case number as a limitation to reliably evaluate the outcomes

Line 456: please add that also different strength MRI scanners affect the interpretation of the imaging findings

Line 457: ‘by specialised radiologists’ – please specify the qualifications

Please discuss why is DWI interpretation to identify infarctions in cases of PA is challenging

Conclusions

Line 468 - It seems to be a slight overstatement that the outcome is defined as good (also please use favourable in the entire text) when 20% of dogs did not survive the first few days after the diagnosis was made. Please rephrase to 'in the majority of canine patients' or similar. 

Lines 468-469: please add the fact that certain amount (almost half?) of dogs have transient (rapidly spontaneously improving/resolving) neurological signs. 

Author Response

REV 1

Dear Reviewer,

the authors would like to thank you for the precious recommendations aimed to improve our manuscript.

Materials and methods were better defined and explained. Furthermore, tables used to show results were totally changed in order to make data more easily available for the reader. In particular, we inserted two specific tables to describe clinical sign’s onset and outcome respectively. A new table was added in the supplementary materials with the detailed neurological exam of each dog. Instead, tables showing CT and MRI features were modified showing main imaging characteristics for each study performed.

Finally, the paper has been revised by a native English speaker in order to improve its form.

We have tried to answer to all the concerns, as detailed below. We hope you will appreciate changes made so that the article can now be considered for publication.

A brief summary 

The aim of the article was to describe the clinical and neurological signs, DI findings and outcome in dogs with imaging diagnosis of PA in a considerable number of dogs. Interresting and useful information is described. In addition, the number of cases for a case series is quite high. Neurology clinicians can find the information from this article useful for reading/interpreting the MRI scans and making the clinical decisions/discussing the prognosis with the pet owners. 

General concept comments

Article

A retrospective study with lacking homogeneous data in order to be able to draw strong consslusions. In addition, the histopathological confirmation is lacking (for very obvious and plausible reasons) and it should be stated that the diagnosis is a suspected diagnosis based on DI findings (or a radiological diagnosis). In addition, it is impossible to say for sure that all neurological signs, especially the transient ones, are attributable to the PA without performing a histopathological examination.

Author: dear Reviewer, thank you for the suggestion. Since we totally agree with you, we decided to change the title of the study using the expression “suspected pituitary apoplexy”. Furthermore, the lack of histopathologic diagnosis was specified and discussed in the study limitations.

A clear description of the diagnostic imaging findings is missing and the reader might be confused how to identify the radiological findings in cases of PA.

Author: dear Reviewer, thank you for the suggestion. The primary objective of this paper is to describe the clinical presentation of dogs with PA, identifying neurological signs that can be associated to PA. The diagnosis of PA, without histopathological confirmation, is based on clinical presentation with the support of advanced imaging techniques. Cases with obvious concomitant intracranial pathologies (such as other intracranial neoplastic lesion or suggestive of meningoencephalitis of unknown origin) were excluded in order to avoid mistakes resulting from the absence of histopathological confirmation.

Associations between the affected pituitary gland findings (hemorrhages, necrosis, ischaemic infacrtions)as well as size of the mass and the history, clinical, neurological signs, and the outcome is missing. If any associations are identified, they will help clinicians to make clinical decision and communicate prognosis with the owners of affected animals. 

Author: thank you for the observation. Unfortunately, a statistical association could not be performed because of statistical limitations due to the low number of included cases, the absence of a control group and of a subpopulation of affected dogs to use for comparison. Furthermore, all dogs included in the study had enlarged pituitary tumours with P:B ratio >0.31. Hence a comparison between enlarged and non-enlarged pituitary tumours could not be made. A statement was added in the discussion.

There is no hypothesis in the article. 

Author: thank you for the remark. The aim of this study is the detailed description of clinical and neurological signs that dogs affected with PA can show. In fact, in veterinary literature a description of the clinical presentation in a consistent number of dogs is lacking.

Specific comments

Abstract

Line 19 - please change 'good' outcome to 'favourable' in the entire text

Author: thank you, we changed it.

Introduction

Line 61: Please provide a short overview of outcomes described so far in dogs suffering from PA in the introduction. It is important for the readers to know that in previously described cases some dogs died during the imaging process or their neurological status deteriorated and therefore they were euthanised. 

Author: thank you for the suggestion. We added a sentence specifying the outcome in previously reported veterinary cases.

M&M

Line 71: what do you mean by ‘collection procedures’? Please specify.

Author: thank you for the question. With “collection procedures” we considered all medical procedures performed in order to make a proper diagnosis at the time of presentation and during hospitalisation. However, this sentence was delated as asked by another Reviewer.

Lines 81-82: only dogs with radiologically confirmed PA were included in the study? Please state it in the text.

Author: thank you for the question. CT and or MRI studies compatible with suspected PA constitute one of inclusion criteria as specified at point 3 (Mat&Met section). Furthermore, in the section of exclusion criteria we explained that cases in which CT/MRI reveal other concomitant intracranial diseases were excluded.

Line 82: please state what was the minimum follow-up time period required to be included in the study. Were the dogs only with long-term follow-up included?

Author: thank you for the question. This point of inclusion criteria was modified according to suggestions of another Reviewer. We delated the point 4 from inclusion criteria and specified time of resolution of clinical signs and follow-up time were recorded only when available. Furthermore, due to the retrospective nature of the study, we did not set a minimum long-term follow-up time. However, the lower long-term follow-up time was 1 month.

Lines 83-84: if phone interview based outcome data collection was unavailable/unsucessful, were these patients still included in the study, please specify.

Author: thank you for the question. We modified inclusion criteria and specified that time of resolution of clinical signs and long-term follow-up time were recorded only when available, using both medical records and phone calls to owners.

Lines 99-103: Please remove the detailed description of neurological examination in the M&M section. The description of neurological examination is provided in separate dedicated textbooks.

Author: thank you for your suggestion. We agree with you and we think it can be redundant. However, since the article is not addressed only to neurologists, we prefer to remind the reader what constitutes a neurological examination.

Lines 129-130: Please describe who were reviewing the CT and MR images - were they in all instances evaluated and reported by a board certified diagnostic imaging specialists?

Author: thank you for the question. We confirm that all CT scans and MRI studies were revised by or Head of Department of Diagnostic Imaging or by board certified diagnostic imaging specialists.

Lines 116-121: Please describe in more details the parameters used to generate the MRI scans and please include the description of DWI parameters, especially the b values chosen for the DWI.

Author: thank you for the suggestion. DWI was acquired with a b value of 0 and 1000. The intensity evaluation was made using b1000 value. We specified this in the main text.

Line 129: do you mean between the two imaging modalities? Please specify.

Author: thank you for the question. We compared information collected from both imaging modalities when dogs underwent both procedures in order to verify if they identified the same lesions.

Lines 137-138: dogs that died during hospitalisation for the causes related to PA should be included in the survival time assessment.

Author: thank you for the suggestion. We modified the calculation of survival time including dogs that died at the time of diagnosis as was suggested also bay another Reviewer.

Results

Lines 148-149: why do you state that dogs were excluded because no clear hemorrhage was identified in the CT scans when PA is an umbrella term not only for hemorrhages? The sentence needs more specifications.

Author: thank you for the suggestion. We changed the sentence in order to specify that in those cases haemorrhagic or infarctual lesion was not identified originating clearly from a pituitary mass.

Lines 151-152: I missed a clear statement in the M&M section inclusion criteria that only dogs with radiological diagnosis of PA in conjunction of abnormal neurological examination/history of recent neurological signs were included. Please specify in the M&M section.

Author: thank you for the observation. In the “materials and methods’” section we specified that dogs without clinical or neurological signs were excluded from the study. The primary objective of this paper is to describe the clinical presentation of dogs with PA, identifying neurological signs that can be associated to PA. In our study, imaging techniques is needed only to support the diagnosis. In this specific case, PA was diagnosed with a CT performed to control an already known pituitary mass. The dog did not show any clinical sign at the time of imaging diagnosis.

Line 155: change to ‘remaining’

Author: thank you for the suggestion. The sentence as changed according to indications of another Reviewer.

Table 1: please specify in the table section long-term follow-up what was the time period of follow-up in each case (e.g., 7 months etc.)

Author: thank you for the suggestion. Long-term follow up was recorded only when available using medical records and phone calls to owners. Unfortunately, due to the retrospective nature of this study, we did not set a minimum long-term follow-up time. However, the lower long-term follow-up time was 1 month.

Line 210-211: change ‘proprioceptive’ deficits to ‘postural reaction’ deficits in the entire text

Author: thank you, we changed it.

Line 249: change ‘sequences’ to ‘images’

Author: thank you, we changed it.

Lines 249-251: the sentence describing signal intensities in MR images needs to be rephrased

Author: thank you, the sentence was rephrased.

Line 263: please specify what was exactly found in the CT images in this case.

Author: thank you for the suggestion. We added a sentence explaining CT findings in this specific case.

Lines 265-266: The P:B ratios are quite variable in case series population. Was the size of the affected pituitary gland/mass associated with the onset, severity and rapidity of resolution of the clinical signs? It might help to create groups of the sizes and investigate these associations

Author: thank you for the suggestion. We wondered the same questions and we would have liked to investigate also this aspect. However, since all dogs had an enlarged pituitary mass (P:B > 0.31) this was not possible. To clarify this concept, we added a sentence in the discussion.

Line 302 - authors state that in 15 dogs the outcome was available. Please specify in results section why follow-up was not available in the rest 4 dogs included in the case series

Author: thank you for the suggestion. We clarified this aspect in the “materials and method’s” section.

 Tables 3 and 4: I do not find this type of data presentation very helpful to the reader. More useful tables would be describing imaging sequence findings in conjunction with each other and in addition, comparison to CT findings in the same cases with inclusion of actual representative images.

Author: thank you for the suggestion. Since we agree with you, we completely modified tables 3 and 4. In new tables 3 and 4 there is the detailed description of CT and MRI findings for each case so that the readers should find information easily.

Please create a table with groups of MRI and CT patterns of changes detected in the pituitary mass

E.g.,

Group 1: 4 cases, suspected acute hemorrhage: T2W SI hypo, T1W SI iso, T2* signal void etc. (in all 4 cases the imaging findings are roughly the same) and pair them with CT findings (where available) as well.

Group 2: 5 cases, suspected acute hemorrhage and infarction: T2WI SI heterogeneous hypo and hyper etc.

Group 3: 2 cases, suspected haemorrhage and necrosis: etc.

Author: thank you for the suggestion. In our opinion data are easily available for consultation using new tables 3 and 4.

Provide typical paired MRI and CT images for each (at least with bigger sample size) group as an example.,

Author: thank you for the suggestion. For the Authors the actual included images are sufficiently explicative. As the primary objective of this study is to give a detailed description of clinical and neurological signs, we do not believe that including images divided in groups would be useful. However, we do thank you for this very interesting advice. We are currently working on a new study specifically focused on MRI and CT findings in dogs with PA.

Please investigate if radiological finding patterns were associated with any clinical data (the rapidity, severity of neurological signs and/or rapidity of resolution of neurological signs and overall outcome)

Author: thank you for the suggestion. We would like to investigate this aspect in another study, trying to obtain more homogeneous data. At present, due to the retrospective nature of this study, data are too variable to allow a correct statistical analysis.

Line 324: what was the median and/or mean and range of follow-up time in dogs with available long term follow-up?

Author: thank you for the suggestion. We added the median long-term follow up time and the relative range.

Discussion

Lines 346-349: sentences need to be rephrased

Author: thank you, we changed it.

Lines 349-353: please divide the long sentence into two sentences with clear meaning. Use term ‘mass effect’ where appropriate

Author: thank you, we changed it.

Line 356 add headache in animals is a subjective finding in many instances, please clarify in the text

Author: thank you, we modified the sentence.

Lines 362 and 364 ‘neuro-ophthalmic’ or ‘neuro-ophthalmological’? please be consistent

Author: thank you, we changed it.

Line 372 please add ‘possible’ or ‘suspected’ vascular nature as there are no histopathological evidence available

Author: thank you, we changed it.

Lines 381-382 please rephrase the sentence

Author: thank you, we changed it.

Lines 390-392: please explain/discuss how the diagnoses of different types of PA made by use of advanced imaging in humans aid in guiding the patient treatment. How does it or does it not translate in canine patients?

Author: thank you for the suggestion. We added a sentence for a better explanation of this concept. In human medicine, MRI using also DWI sequences represents a fundamental tool for a prompt diagnosis and subsequent therapeutical decisions. In veterinary medicine, a more standardized therapeutical protocol is still lacking. However, making a precise diagnosis would help the clinician in the decision making process.

Line 397: did the size of the pituitary tumour were associated with the rapidity, severity of neurological signs and/or rapidity of resolution of neurological signs and overall outcome? Please discuss and add these findings into result section.

Author: thank you for the suggestion. We wondered the same questions and we would have liked to investigate also this aspect. However, since all dogs had an enlarged pituitary mass (P:B > 0.31) this was not possible. To clarify this concept, we added a sentence in the discussion.

Line 455: please add low case number as a limitation to reliably evaluate the outcomes

Author: thank you, we added it.

Line 456: please add that also different strength MRI scanners affect the interpretation of the imaging findings

Author: thank you, we added it.

Line 457: ‘by specialised radiologists’ – please specify the qualifications

Author: thank you for the suggestion. According to another reviewer we modified the sentence. All CT scans and MRI studies were revised by or Head of Department of Diagnostic Imaging or by board certified diagnostic imaging specialists.

Please discuss why is DWI interpretation to identify infarctions in cases of PA is challenging

Author: thank you for the suggestion. We tried to give a better description of the utility of this sequence to the reader in the discussion section. 

Conclusions

Line 468 - It seems to be a slight overstatement that the outcome is defined as good (also please use favourable in the entire text) when 20% of dogs did not survive the first few days after the diagnosis was made. Please rephrase to 'in the majority of canine patients' or similar. 

Author: thank you for the observation. We rephrased the sentence and focused more specifically on recovery time from onset clinical signs.

Lines 468-469: please add the fact that certain amount (almost half?) of dogs have transient (rapidly spontaneously improving/resolving) neurological signs.

Author: thank you for the suggestion, we added it.

Reviewer 2 Report

Dear authors,

Thank you for submitting an interesting retrospective study regarding canine pituitary apoplexy (PA). Pituitary apoplexy has traditionally considered a poor prognosis entity however your results might indicate otherwise.

PA is currently understood as a syndrome characterised by acute neurologic signs related to sudden compression of parasellar structures, usually due to haemorrhagic infarction of a pituitary tumour. It would be interesting to know if any of the cases included in this article showed an ischaemic infarct as otherwise your discussion should be based on that as well as the previously reported cases in the veterinary literature that were also associated to a haemorrhagic infarction.

There were several places where the subject-verb agreement was incorrect and a few sentences that I had trouble following. I think detailed copy editing would be helpful.

Some of the authors were included in previous reports of PA, therefore it is strictly recommended to mention if some of cases were already reported somewhere else.

I believe it is an interesting article but there are several points should be first addressed before considering it for publication.

ABSTRACT

Line 9 - Please amend characterized for characterised.

Line 10 - Has pituitary apoplexy ever been reported in a normal canine or feline pituitary gland? If not please, amend the sentence.

Line 12 - Please delete in detail.

Line 12 - Please amend neurological abnormalities for abnormalities in the neurological examination.

Line 15 – please add suspected - Nineteen cases of suspected PA were included

Line 15 - Blindness? The diagnosis is usually performed combining neurological signs and imaging findings?

Line 17 - Onset?

Line 19-20 - Good prognosis - median survival time of 7.5 months from the time of PA diagnosis.

Line 22-23 - A CT and/or MRI investigation is fundamental to promptly diagnose PA and differentiate 22 acute/chronic haemorrhages from necrosis and cystic components.

INTRODUCTION

Line 31 – Please explain that PA has not been reported in a normal canine or feline pituitary gland.

Line 35-41 – How would that theory explain PA in a normal healthy pituitary gland?

Line 41 – occurrences 

Line 50-52 – Please amend - but can also resolve with conservative treatment after a longer period of time.

Line 54 – please amend - in a few dogs….and ophthalmoplegia

Line 57 - Multidetector-row computed tomography (MDCT)

Line 58 – please amend - …with histopathological confirmation.

Line 62 – please delete detailed. 

Line 63 – please add …advanced imaging findings…

Line 63-64 – Please amend the sentence to… outcome of a large number of dogs with suspected PA. 

Based exclusively on the clinical signs reported in your cases, PA cannot be suspected, especially if no previous confirmation of a pituitary mass has been reported. Histopathology would confirm the final diagnosis while advanced imaging would make it most likely.

MATERIAL AND METHODS

Was any of the cases included in this study previously reported in any other scientific article? If so, please explain.

Line 69 – The study population included client-owned dogs with a history of acute onset of neurological signs but the abstract states that only in 73.4% of the cases the onset of the neurological signs was acute. Please explain the discrepancy.

Line 71-72 – Please delete “All collection procedures were performed solely for the dog’s benefit and for standard diagnostic and monitoring purposes”

The inclusion and exclusion criteria need to be written. PA cannot be suspected based exclusively on the onset and the neurological examination as many other different diseases can cause equal or similar clinical signs as the ones reported in this article. 
MDCT scans and/or MRI study of the head compatible with suspected PA should be included as an inclusion criteria.

As exclusion criteria, please explain which neoplasia were excluded as PA is classically associated to a pituitary tumour.

CLINICAL AND NEUROLOGICAL EVALUATION

Line 94-95 - …and any previously diagnosis of a pituitary mass was recorded.

Line 96-98 – please amend - The onset of clinical signs, before presentation, was defined as per-acute (when clinical signs developed in less than 24 h), acute (1-7 days) or chronic (over weeks or months).

Line 99-101 – The animal’s behaviour and the presence of epileptic seizures are cannot be included within the neurological examination as they are clinical signs. Please re-write the sentence.

Line 102- Please delete pain. Amend – localisation.
DIAGNOSTIC IMAGING FINDINGS

Line 113/115 – please amend analysed

Line 199 – Was apparent diffusion coefficients (ADC) done with the low field MRI-scans as well? If not please explain.

Line 120 – unless stated otherwise.

Line 123 – The presence of oedema is not always associated to a mass effect so please amend.

Line 129 – Please amend - Finally, agreement between the two procedures was reported considering conclusions made on final reports.

OUTCOME

Line 132 – Treatments were recorded including the ones in the emergency setting.

Line 133 – The time for the neurological sign’s resolution was recorded once available and classified as…

Line 136 - Long term follow-up was recorded using…

Line 137 – euthanised

Line 138 – hospitalisation / one month

Line 139 – between one and six months; between six months and one year and more than one year.

Line 139-140 – why were euthanised dogs or spontaneously dead during hospitalization excluded from the survival time?

Line 141 – or due to an unrelated cause.

RESULTS

CASE POPULATION

Since the study only includes 19 dogs there is no need to write percentages so please delete them all in this section.

Line 147 – From medical record’s review…

Line 147 - 24 cases were initially included for data analysis.

Line 150-151 – These two dogs were also excluded.

Line 151-153 – Please explain this case. Was the dog diagnosed with a PA but he did not show any clinical or neurological abnormalities? According to the inclusion criteria, a normal neurological examination or lack of neurological signs are not a reason to be excluded.

Line 154 – 155 – The breed distribution was mixed-breed (9/19), Labrador retrievers…..

Table 1 – Please delete detailed.

Table 1 – The neurological examination and the neurological signs should be reported individually; therefore, behavioural changes should not be included within the findings of the neurological examination or the onset. Please add a new column for the neurological signs.

-    Case 10,14, 15, 16 and 19 do not have any follow-up as stated in the inclusion criteria. 
-    Please add Death in the long-term follow up for cases 4, 6, 8, 9, 11, 12, 13, 17 and 18

To be able to include cases without a follow-up, the inclusion criteria would need to be changed.

ONSET OF CLINICAL SIGNS and NEUROLOGICAL FINDINGS

These two sections need to be written. Neurological signs and findings in the neurological examinations are mixed and should be explained individually.

I would recommend to write a section of clinical signs with a subdivision for onset and a section for the neurological examination.

Since the study only includes 19 dogs there is no need to write percentages so please delete them all in these two sections.

Figure 1 should be deleted as it is not relevant for the study.

Table 2 – Please delete behaviour.

IMAGING FINDINGS

Since the study only includes 19 dogs there is no need to write percentages so please delete them all in this section.

For example, in line 239-40 - Eighteen out of 19 dogs underwent MDCT examination. Nine dogs underwent MRI. Eight dogs underwent both procedures. In all cases a pituitary mass was identified.

Line 252 – Please state if the DWI and ADC maps were performed only with the high field MRI.

Line 258 -261 – Based on that sentence, it seems like those 4 dogs did not necessarily have a PA as necrosis can happen within a tumour and it is not classified as PA. Did those 4 cases have acute or chronic onset?

Figure 4 – Which case number does this MRI belong to? I would struggle to believe that dog showed any clinical signs.

TREATMENT AND OUTCOME

Since the study only includes 19 dogs there is no need to write percentages so please delete them all in this section.

Line 297 - Nine dogs received some medical treatment.

Line 301 - …in addition to the medical therapy.

Line 302 - If the follow-up was only available in 15 cases that should not be an inclusion criteria. 
Neurological sign’s resolution.

Line 306 - …resolution of the neurological signs…

Line 313 – One dog with behavioural changes….

Line 316 – hospitalisation

Line 319 – The remaining dog was euthanised….

Lines 325/328 – euthanised

DISCUSSION

Line 337 - …suggestive of haemorrhage or infarction….

Line 343 - …epileptic seizures…

Line 345 – characterised by neurological signs. Please delete symptoms.

Line 347 – Delete sign.

Line 348 – Behavioural changes was the most common clinical sign found in this study as previously reported.

Line 352 - …as it can happen in a case of PA….
Did any of your cases or the previously reported cases have an ischaemic lesion? If not, I would recommend to just mention haemorrhage, as an ischemic lesion is less likely to lead to a sudden increase of ICP

Line 357 – …base of the head, secondary to….

Line 360-61 – Please delete supposed to reflect.

Line 362 – 364 - In the present study, cranial nerve examination revealed only neuro-ophthalmological deficits. That sentence does not make sense.

Line 365 – In dogs, the hypophyseal….

It would have been interesting to discuss why some of the other cases had visual deficits compared to your cases. Tumour/haemorrhage size?

Line 372 - In this study, the neurological signs resolved in less than 24 hours in the majority 
cases and most of them showed vestibular signs. That sentence does not make sense!!
Please re-write the following sentence - This reflects the vascular nature of PA which, as other more common vascular lesions such as transient ischaemic attacks, has the potential for a fast-clinical resolution.
The previously reported cases showed a haemorrhagic lesion which would not improve within 24 hours and TIA do not show on an MRI-scan, hence the rapid resolution.

Lines 380 – 392– Needs to be re-written. Make clear statements in regards to a haemorrhagic or an ischaemic lesion.

Line 397 – Was the P:B ratio > 0.31 in the case showed in picture 4.

Line 403-406 – Has any specific treatment been identified for PA? How common is a cavernous sinus thrombosis?
This sentence applies to human medicine so please delete.

Line 411 – If the lesion was necrotic, would it be classified as PA?

Line 424 – 430 – Please delete this paragraph as no relevant. It is generally assumed that blood degradation products in intracerebral hematomas are composed primarily of oxyhemoglobin (oxyHb) in the hyperacute stage (<24 hours), deoxyhemoglobin (deoxyHb) in the acute stage (1–3 days), intracellular methemoglobin (metHb) in the early subacute stage (3–7days), extracellular metHb in the late subacute stage (7–14days), and hemosiderin in the chronic stage (>14 days) [39]. During blood degradation, the molar magnetic susceptibility increases, from slightly negative relative to cerebrospinal fluid (CSF) in oxyHb to highly positive in deoxyHb, metHb, and hemosiderin [39].

Line 436-437 - Not surprisingly, the outcome has revealed to be overall good, with the vast majority of dogs surviving acute symptoms and living for months or even years after the diagnosis of PA. Is a median survival time of 7.5 months a good outcome? Maybe PA has a good short-term outcome but a better explanation is needed for a long-term outcome. Please re-write that sentence.

Line 437 – This is similar to human medicine, where the outcome is generally good.

Line 452 - The patient should be given some time to recover from the vascular event, considering euthanasia in cases where there is no recovery and severe neurological signs.

Line 457 – evaluated by imaging specialists.

Author Response

REV 2

Dear Reviewer,

the authors would like to thank you for the precious recommendations aimed to improve our manuscript.

Each section of the article has been revised in detail with the aim of improving the comprehension and the design of the study. In particular, materials and methods were partially reformulated and the result’s section was better summarized. Furthermore, tables used to show results were totally changed in order to make data more easily available for the reader. In particular, we inserted two specific tables to describe clinical sign’s onset and outcome respectively. A new table was added in the supplementary materials with the detailed neurological exam of each dog. Instead, tables showing CT and MRI features were modified showing main imaging characteristics for each study performed.

Finally, the paper has been extensively revised by a native English speaker in order to improve its form.

We have tried to answer to all the concerns, as detailed below. We hope you will appreciate changes made so that the article can now be considered for publication.

Dear authors,

Thank you for submitting an interesting retrospective study regarding canine pituitary apoplexy (PA). Pituitary apoplexy has traditionally considered a poor prognosis entity however your results might indicate otherwise.

PA is currently understood as a syndrome characterised by acute neurologic signs related to sudden compression of parasellar structures, usually due to haemorrhagic infarction of a pituitary tumour. It would be interesting to know if any of the cases included in this article showed an ischaemic infarct as otherwise your discussion should be based on that as well as the previously reported cases in the veterinary literature that were also associated to a haemorrhagic infarction.

Author: dear Reviewer, thank you for the suggestion. In this study all identified pituitary lesions had characteristics suggestive of haemorrhagic infarction. No evident signal alterations indicating ischaemic lesions were identified. Hence, specific test modifications have been made along the whole text.

There were several places where the subject-verb agreement was incorrect and a few sentences that I had trouble following. I think detailed copy editing would be helpful.

Author: dear Reviewer, thank you for the suggestion. We submitted the paper to an English mother tongue in order to ameliorate the whole manuscript.

Some of the authors were included in previous reports of PA, therefore it is strictly recommended to mention if some of cases were already reported somewhere else.

Author: dear Reviewer, thank you for the suggestion. Of our 19 cases only two have been previously described. In particular, dog #12 have been previously described by Briola et al., in 2020 and dog #19 was described by Bertolini et al. in 2007. We are sorry for the missing information in the text and we added this information in the Mat&Met section as requested.

I believe it is an interesting article but there are several points should be first addressed before considering it for publication.

ABSTRACT

Line 9 - Please amend characterized for characterised.

Author: dear Reviewer, thank you. We changed it.

Line 10 - Has pituitary apoplexy ever been reported in a normal canine or feline pituitary gland? If not please, amend the sentence.

Author: dear Reviewer, thank you for the question. This refers to the definition of pituitary apoplexy commonly used in human medicine. We have now specified it in the sentence, hoping it can be more clear.

Line 12 - Please delete in detail.

Author: thank you, we changed it.

Line 12 - Please amend neurological abnormalities for abnormalities in the neurological examination.

Author: thank you, we changed it.

Line 15 – please add suspected - Nineteen cases of suspected PA were included

Author: dear Reviewer, thank you for the suggestion. We decided to change it also in the title.

Line 15 - Blindness? The diagnosis is usually performed combining neurological signs and imaging findings?

Author: dear Reviewer, thank you for the question. The occurrence of blindness and deficits at the neuro-ophthalmological examination were described in detail in the main text. We did not consider these findings relevant for the abstract as it occurred in a low number of cases. We confirm that PA diagnosis has been performed combining neurological signs and imaging findings. However, we did not specify this in the abstract in order not to exceed the maximum number of words. For a better understanding, we detailed it in the “materials and methods” description.

Line 17 - Onset?

Author: thank you, we changed it.

Line 19-20 - Good prognosis - median survival time of 7.5 months from the time of PA diagnosis.

Author: dear Reviewer, thank you for the suggestion. As this data was confusing, we decided to describe the prognosis as “favourable” pointing that as a “short-term prognosis”. The term “favourable” was suggested by the other Reviewer.

Line 22-23 - A CT and/or MRI investigation is fundamental to promptly diagnose PA and differentiate 22 acute/chronic haemorrhages from necrosis and cystic components.

Author: dear Reviewer, thank you for the remark. We removed this sentence from the text.

INTRODUCTION

Line 31 – Please explain that PA has not been reported in a normal canine or feline pituitary gland.

Author: dear Reviewer, thank you for the observation. We added a sentence in the text to specify that.

Line 35-41 – How would that theory explain PA in a normal healthy pituitary gland?

Author: dear Reviewer, thank you for the question. The paragraph has been modified accordingly. Reports of PA in a normal pituitary gland are rare also in human literature and its pathophysiology is supposed to be different but is not clearly understood.

Line 41 – occurrences 

Author: thank you, we changed it.

Line 50-52 – Please amend - but can also resolve with conservative treatment after a longer period of time.

Author: thank you, we changed it.

Line 54 – please amend - in a few dogs….and ophthalmoplegia

Author: thank you, we changed it.

Line 57 - Multidetector-row computed tomography (MDCT)

Author: thank you for the remark. We changed it in “Computed tomography (CT)”.

Line 58 – please amend - …with histopathological confirmation.

Author: thank you, we changed it.

Line 62 – please delete detailed. 

Author: thank you, we changed it.

Line 63 – please add …advanced imaging findings…

Author: thank you, we changed it.

Line 63-64 – Please amend the sentence to… outcome of a large number of dogs with suspected PA. 

Author: thank you, we changed it.

Based exclusively on the clinical signs reported in your cases, PA cannot be suspected, especially if no previous confirmation of a pituitary mass has been reported. Histopathology would confirm the final diagnosis while advanced imaging would make it most likely.

Author: dear Reviewer, thank you for the suggestion. We totally agree with your remark and for this reason we removed the word “clinically” from the sentence and revised the statement as you suggested.

MATERIAL AND METHODS

Was any of the cases included in this study previously reported in any other scientific article? If so, please explain.

Author: dear Reviewer, thank you for the suggestion. Of our 19 cases only two have been previously described. In particular, dog #12 have been previously described by Briola et al., in 2020 and dog #19 … was described by Bertolini et al. in 2007. We are sorry for the missing information in the text and we added this information in the Mat&Met section as requested.

Line 69 – The study population included client-owned dogs with a history of acute onset of neurological signs but the abstract states that only in 73.4% of the cases the onset of the neurological signs was acute. Please explain the discrepancy.

Author: thank you for the observation. We modified the sentence.

Line 71-72 – Please delete “All collection procedures were performed solely for the dog’s benefit and for standard diagnostic and monitoring purposes”

Author: thank you for the suggestion. The sentence has been delated.

The inclusion and exclusion criteria need to be written. PA cannot be suspected based exclusively on the onset and the neurological examination as many other different diseases can cause equal or similar clinical signs as the ones reported in this article. 
MDCT scans and/or MRI study of the head compatible with suspected PA should be included as an inclusion criteria.

Author: dear Reviewer, thank you for the suggestion. We specified more clearly that MRI and CT images needed to be suggestive of a suspected PA in order to be considered for inclusion.

As exclusion criteria, please explain which neoplasia were excluded as PA is classically associated to a pituitary tumour.

Author: thank you for the observation. We specified that cases with neoplasia other than the pituitary tumour were excluded.

CLINICAL AND NEUROLOGICAL EVALUATION

Line 94-95 - …and any previously diagnosis of a pituitary mass was recorded.

Author: thank you, we changed it.

Line 96-98 – please amend - The onset of clinical signs, before presentation, was defined as per-acute (when clinical signs developed in less than 24 h), acute (1-7 days) or chronic (over weeks or months).

Author: thank you for the suggestion. We changed the onset classification.

Line 99-101 – The animal’s behaviour and the presence of epileptic seizures are cannot be included within the neurological examination as they are clinical signs. Please re-write the sentence.

Author: thank you for the suggestion. We separated epileptic seizures from the neurological signs. However, we would like to specify that “mental status and behaviour” constitute a standard part of the neurological examination.

Line 102- Please delete pain. Amend – localisation.

Author: thank you, we changed it.

DIAGNOSTIC IMAGING FINDINGS

Line 113/115 – please amend analysed

Author: thank you, we changed it.

Line 199 – Was apparent diffusion coefficients (ADC) done with the low field MRI-scans as well? If not please explain.

Author: thank you for the question. Diffusion weighted imaging (DWI) and apparent diffusion coefficients (ADC) were performed only using high field MRI since protocols applied with the previous low field MRI were not intended for the acquisition of DWI sequences and ADC maps. This has been specified in the text.

Line 120 – unless stated otherwise.

Author: thank you, we changed it.

Line 123 – The presence of oedema is not always associated to a mass effect so please amend.

Author: thank you for the suggestion. We changed it.

Line 129 – Please amend - Finally, agreement between the two procedures was reported considering conclusions made on final reports.

Author: thank you, we changed it.

OUTCOME

Line 132 – Treatments were recorded including the ones in the emergency setting.

Author: thank you, we changed it.

Line 133 – The time for the neurological sign’s resolution was recorded once available and classified as…

Author: thank you, we changed it.

Line 136 - Long term follow-up was recorded using…

Author: thank you, we changed it.

Line 137 – euthanized

Author: thank you, we changed the sentence.

Line 138 – hospitalisation / one month

Author: thank you, we changed it.

Line 139 – between one and six months; between six months and one year and more than one year.

Author: thank you, we changed it.

Line 139-140 – why were euthanised dogs or spontaneously dead during hospitalization excluded from the survival time?

Author: thank you for the suggestion. We included dogs euthanised or spontaneously dead during hospitalization in the survival time assessment. We also modified inclusion criteria, specifying that the time of recovery from clinical signs and long-term follow up were collected only when available, using both medical records and phone calls to owners.

Line 141 – or due to an unrelated cause.

Author: thank you, we changed it.

RESULTS

CASE POPULATION

Since the study only includes 19 dogs there is no need to write percentages so please delete them all in this section.

Author: thank you, we delated it.

Line 147 – From medical record’s review…

Author: thank you, we changed it.

Line 147 - 24 cases were initially included for data analysis.

Author: thank you, we changed it.

Line 150-151 – These two dogs were also excluded.

Author: thank you, we changed it.

Line 151-153 – Please explain this case. Was the dog diagnosed with a PA but he did not show any clinical or neurological abnormalities? According to the inclusion criteria, a normal neurological examination or lack of neurological signs are not a reason to be excluded.

Author: thank you for your observation. We specified in the exclusion criteria that cases without any clinical or neurological sign were excluded. The primary aim of this paper is to describe the clinical presentation of dogs with PA, identifying neurological signs that can be associated to PA. In our study, imaging techniques is needed only to support the diagnosis. In this specific case, PA was diagnosed with a CT performed to control an already known pituitary mass. The dog did not show any clinical sign at the time of imaging diagnosis. Hence, it was excluded.

Line 154 – 155 – The breed distribution was mixed-breed (9/19), Labrador retrievers…..

Author: thank you, we changed it.

Table 1 – Please delete detailed.

Author: thank you, we changed it. Table 1 has been modified focusing on resolution of clinical signs and long-term follow-up (has become Table 5).

Table 1 – The neurological examination and the neurological signs should be reported individually; therefore, behavioural changes should not be included within the findings of the neurological examination or the onset. Please add a new column for the neurological signs.

Author: thank you for your suggestion. We totally agree with you, which is why the neurological examination is described in detail in Table 2. We modified Table 1 (now table 5) with information regarding the recovery and follow up time (when available), diagnostic imaging techniques performed, P/B ratio and mass effect. We created a new Table (Table 1) highlighting aspects of clinical sign’s onset. However, we would like to specify that “mental status and behaviour” constitute a standard part of the neurological examination. Therefore, they were included in the detailed neurological examination, but are included also in the onset because owners often report behavioural changes as a main complain.

-    Case 10,14, 15, 16 and 19 do not have any follow-up as stated in the inclusion criteria. 

Author: thank you for the suggestion. Probably Table 1 was a little bit confusing. We changed the Tables’ information (see Table 5). We also modified the inclusion criteria, specifying that the time of resolution of clinical signs and long-term follow up were collected only when available, using both medical records and phone calls to the owners. Unfortunately, a few owners were not available for the telephone calls or have been changed their phone numbers and/or email address. That’s why we did not have all the follow-ups.

-    Please add Death in the long-term follow up for cases 4, 6, 8, 9, 11, 12, 13, 17 and 18

Author: thank you, we changed it. In order to make information more easily available, we also modified table 1 focusing on time of recovery and follow up when available (see table 5), P/B ratio and mass effect. We created a new table (see new table 1) concentrating on onset characteristics and concomitant signs.

To be able to include cases without a follow-up, the inclusion criteria would need to be changed.

Author: thank you for your observation. We modified inclusion criteria specifying that the time of resolution of clinical signs and long-term follow up were collected only when available, using both medical records and phone calls to the owners. These criteria were better specified in the Mat&Met section.

ONSET OF CLINICAL SIGNS and NEUROLOGICAL FINDINGS

These two sections need to be written. Neurological signs and findings in the neurological examinations are mixed and should be explained individually. I would recommend to write a section of clinical signs with a subdivision for onset and a section for the neurological examination.

Author: thank you for the suggestion. In this section we would like to highlight the onset (type of onset, owners’ main complain and accessory signs). In order to make the reading easier, we created a new table (new table 1). We also modified the paragraph’s text.

Since the study only includes 19 dogs there is no need to write percentages so please delete them all in these two sections.

Author: thank you for the suggestion, we deleted percentages in these sections.

Figure 1 should be deleted as it is not relevant for the study.

Author: thank you for the suggestion.

Table 2 – Please delete behaviour.

Author: dear Reviewer, the Authors prefer to maintain the “behaviour” column as “mental status and behaviour” is a part of the neurological examination. In this table we preferred to separate these two in order to make it easier for the reader to understand the results.

IMAGING FINDINGS

Since the study only includes 19 dogs there is no need to write percentages so please delete them all in this section.

Author: thank you for the suggestion, we deleted percentages in this section.

For example, in line 239-40 - Eighteen out of 19 dogs underwent MDCT examination. Nine dogs underwent MRI. Eight dogs underwent both procedures. In all cases a pituitary mass was identified.

Author: thank you for the suggestion. We delated percentage from this sentence.

Line 252 – Please state if the DWI and ADC maps were performed only with the high field MRI.

Author: thank you for the remark. We added the information.

Line 258 -261 – Based on that sentence, it seems like those 4 dogs did not necessarily have a PA as necrosis can happen within a tumour and it is not classified as PA. Did those 4 cases have acute or chronic onset?

Author: thank you for the question. All these 4 cases had per-acute onset of clinical signs and imaging findings were consistent with haemorrhage or ischaemic necrosis within the pituitary mass using standard sequences. One of the theories for PA pathogenesis supposes that a growing pituitary tumour may outstrip its blood supply resulting in acute ischaemic necrosis and infarction (Bertolini et al., 2007). In order to avoid further doubts we specified that necrosis identified was an ischemic necrosis.

Figure 4 – Which case number does this MRI belong to? I would struggle to believe that dog showed any clinical signs.

Author: thank you for the question. These MRI images belong to case number 2. This dog had per-acute onset of balance loss episodes and pica. The P:B ratio is 0.35.

TREATMENT AND OUTCOME

Since the study only includes 19 dogs there is no need to write percentages so please delete them all in this section.

Author: thank you for the suggestion, we deleted percentages in this section.

Line 297 - Nine dogs received some medical treatment.

Author: thank you, we changed it.

Line 301 - …in addition to the medical therapy.

Author: thank you, we changed it.

Line 302 - If the follow-up was only available in 15 cases that should not be an inclusion criteria. 
Neurological sign’s resolution.

Author: thank you for the suggestion. We modified the inclusion criteria, specifying that the time of resolution of clinical signs and long-term follow up where collected only when available, using both medical records and phone calls to the owners.

Line 306 - …resolution of the neurological signs…

Author: thank you, we changed it.

Line 313 – One dog with behavioural changes….

Author: thank you, we changed it.

Line 316 – hospitalization

Author: thank you, we changed it.

Line 319 – The remaining dog was euthanised….

Author: thank you, we changed it.

Lines 325/328 – euthanized

Author: thank you, we changed it.

DISCUSSION

Line 337 - …suggestive of haemorrhage or infarction….

Author: thank you, we changed it.

Line 343 - …epileptic seizures…

Author: thank you, we changed it.

Line 345 – characterised by neurological signs. Please delete symptoms.

Author: thank you, we changed it.

Line 347 – Delete sign.

Author: thank you, we changed it.

Line 348 – Behavioural changes was the most common clinical sign found in this study as previously reported.

Author: thank you, we changed it.

Line 352 - …as it can happen in a case of PA….
Did any of your cases or the previously reported cases have an ischaemic lesion? If not, I would recommend to just mention haemorrhage, as an ischemic lesion is less likely to lead to a sudden increase of ICP

Author: thank you for the suggestion. We changed it.

Line 357 – …base of the head, secondary to….

Author: thank you, we changed it.

Line 360-61 – Please delete supposed to reflect.

Author: thank you, we changed it.

Line 362 – 364 - In the present study, cranial nerve examination revealed only neuro-ophthalmological deficits. That sentence does not make sense.

Author: thank you for the suggestion. We changed the sentence.

Line 365 – In dogs, the hypophyseal….

Author: thank you, we changed it.

It would have been interesting to discuss why some of the other cases had visual deficits compared to your cases. Tumour/haemorrhage size?

Author: thank you for the question. We added a sentence to explain that this difference can be due to the size of the original tumour or to the haemorrhage extent.

Line 372 - In this study, the neurological signs resolved in less than 24 hours in the majority 
cases and most of them showed vestibular signs. That sentence does not make sense!!
Please re-write the following sentence - This reflects the vascular nature of PA which, as other more common vascular lesions such as transient ischaemic attacks, has the potential for a fast-clinical resolution.
The previously reported cases showed a haemorrhagic lesion which would not improve within 24 hours and TIA do not show on an MRI-scan, hence the rapid resolution.

Author: thank you for the suggestion. As the previous statement was misleading, the authors decided to remove it.

Lines 380 – 392– Needs to be re-written. Make clear statements in regards to a haemorrhagic or an ischaemic lesion.

Author: thank you for the suggestion. We modified the paragraph in order to better clarify the distinction between sequences useful for the recognition of haemorrhagic foci and ischaemic lesions.

Line 397 – Was the P:B ratio > 0.31 in the case showed in picture 4.

Author: thank you for the question. The P:B ratio of case #2 is 0.35.

Line 403-406 – Has any specific treatment been identified for PA? How common is a cavernous sinus thrombosis?
This sentence applies to human medicine so please delete.

Author: thank you for the suggestion. This sentence was referred to human medicine. In order to be more clear we decided to remove it.

Line 411 – If the lesion was necrotic, would it be classified as PA?

Author: thank you for the question. Hyperattenuating areas visualized in CT are suggestive of haemorrhagic foci. Instead, inhomogeneous post-contrast enhancement with hypovascular areas are suggestive of necrotic foci. To the authors knowledge, the growing pituitary tumour may overtake blood supply resulting in acute ischaemic necrosis and infarction.

Line 424 – 430 – Please delete this paragraph as no relevant. It is generally assumed that blood degradation products in intracerebral hematomas are composed primarily of oxyhemoglobin (oxyHb) in the hyperacute stage (<24 hours), deoxyhemoglobin (deoxyHb) in the acute stage (1–3 days), intracellular methemoglobin (metHb) in the early subacute stage (3–7days), extracellular metHb in the late subacute stage (7–14days), and hemosiderin in the chronic stage (>14 days) [39]. During blood degradation, the molar magnetic susceptibility increases, from slightly negative relative to cerebrospinal fluid (CSF) in oxyHb to highly positive in deoxyHb, metHb, and hemosiderin [39].

Author: thank you for the suggestion. We deleted it.

Line 436-437 - Not surprisingly, the outcome has revealed to be overall good, with the vast majority of dogs surviving acute symptoms and living for months or even years after the diagnosis of PA. Is a median survival time of 7.5 months a good outcome? Maybe PA has a good short-term outcome but a better explanation is needed for a long-term outcome. Please re-write that sentence.

Author: thank you for the suggestion. We modified the sentence focusing on recovery time from the occurrence of per-acute and acute signs.

Line 437 – This is similar to human medicine, where the outcome is generally good.

Author: thank you, we changed it.

Line 452 - The patient should be given some time to recover from the vascular event, considering euthanasia in cases where there is no recovery and severe neurological signs.

Author: thank you, we changed it.

Line 457 – evaluated by imaging specialists.

Author: thank you, we changed it.

Reviewer 3 Report

The article demonstrates a methodical study of intracranial lesions that settle in the sella turcica over many years. I congratulate you for the patience you have had to compile all these cases and shed more light on this type of lesions. 
Just a few nuances that could improve your article. 
Because they use corticosteroids in these patients for medical treatment and given the good prognosis in many patients. The differential diagnosis should include MUO, particularly GME, even though it has a low probability given the shape, location and description of the lesion. This is true as long as there is no histopathology diagnosis, so it should be included as a presuntive diagnosis/ possible etiology in these cases as long as there is no definitive histologic diagnosis.
Please, include the differential diagnosis of all possible lesions that sit on the sella turcica toto bring the clinician closer to this type of lesions. 
At least in my PDF the tables show many flaws that prevent correct reading. Please revise them, or consider another way of presenting the data that is more visual and appealing to the reader. 

Author Response

REV 3

Dear Reviewer,

the authors would like to thank you for the precious recommendations aimed to improve our manuscript.

Each section of the article has been revised in detail with the aim of improving the comprehension and the design of the study. We also gave more importance to study limitations in the discussion.

Finally, the paper has been extensively revised by a native English speaker in order to improve its form.

We have tried to answer to all the concerns, as detailed below. We hope you will appreciate changes made so that the article can now be considered for publication

The article demonstrates a methodical study of intracranial lesions that settle in the sella turcica over many years. I congratulate you for the patience you have had to compile all these cases and shed more light on this type of lesions. 
Just a few nuances that could improve your article. 

Because they use corticosteroids in these patients for medical treatment and given the good prognosis in many patients. The differential diagnosis should include MUO, particularly GME, even though it has a low probability given the shape, location and description of the lesion. This is true as long as there is no histopathology diagnosis, so it should be included as a presuntive diagnosis/ possible etiology in these cases as long as there is no definitive histologic diagnosis.

Please, include the differential diagnosis of all possible lesions that sit on the sella turcica toto bring the clinician closer to this type of lesions. 

Author: dear Reviewer, thank you for the suggestion. In the discussion’s section we added a small paragraph to specify that lesions of other nature would have had a different aspect even if a histopathological definitive diagnosis is lacking due to the favourable outcome of PA in our study. Furthermore, many dogs were already diagnosed with a pituitary mass before presentation for PA.

At least in my PDF the tables show many flaws that prevent correct reading. Please revise them, or consider another way of presenting the data that is more visual and appealing to the reader. 

Author: dear Reviewer, thank you for the suggestion. Tables were completely modified and revised in order to make information more easily available for the reader.

Round 2

Reviewer 1 Report

Dear Authors,

The article has improved significantly.

It is an unfortunate that no associations could be identified between the imaging findings/size groups of the affected pituitary gland and the onset of clinical signs, clinical signs itself and the outcome. It would have been very useful to the readers to have this information to help make clinical decisions. However, l understand that limited case number is precluding these investigations. 

In addition, some English language style editing is still needed as it will help the reader to understand the concept of the article better.

Very few things that I suggest to change are the following:

I suggest to include a column in the tables 3 and 4 with the radiological diagnosis.

Lines 99-103: Please remove the detailed description of neurological examination in the M&M section. The description of neurological examination is provided in separate dedicated textbooks.

Author: thank you for your suggestion. We agree with you and we think it can be redundant. However, since the article is not addressed only to neurologists, we prefer to remind the reader what constitutes a neurological examination.

If you prefer to keep the description of the neurological examination in the article, please add a table describing the neurological examination rather than describing it in a text.

Lines 129-130: Please describe who were reviewing the CT and MR images - were they in all instances evaluated and reported by a board certified diagnostic imaging specialists?

Author: thank you for the question. We confirm that all CT scans and MRI studies were revised by or Head of Department of Diagnostic Imaging or by board certified diagnostic imaging specialists.

What is the degree/undergone training/affiliations (board certified?) of the head of the DI department? Please include a more detailed description in the text. Please add the specification if the diagnostic imaging specialists interpreting the images were board certified. The same applies to neurologists involved in the study.

Table 1: please specify in the table section long-term follow-up what was the time period of follow-up in each case (e.g., 7 months etc.)

Author: thank you for the suggestion. Long-term follow up was recorded only when available using medical records and phone calls to owners. Unfortunately, due to the retrospective nature of this study, we did not set a minimum long-term follow-up time. However, the lower long-term follow-up time was 1 month.

Please include in the table section describing the long-term follow-up in cases where it says the animal was ‘alive’ specification for how long after the diagnosis the animal survived until the study was performed.

Author Response

REV1

Dear Reviewer,

The authors would like to thank you once again for the useful suggestions.

Tables were improved in order to make it easier for the reader to understand results. Information concerning author’s degrees were added.

Finally, the paper has been extensively revised by a native English speaker in order to improve its form.

We hope you will appreciate changes made so that the article can now be considered for publication.

Dear Authors,

The article has improved significantly.

It is an unfortunate that no associations could be identified between the imaging findings/size groups of the affected pituitary gland and the onset of clinical signs, clinical signs itself and the outcome. It would have been very useful to the readers to have this information to help make clinical decisions. However, l understand that limited case number is precluding these investigations. 

In addition, some English language style editing is still needed as it will help the reader to understand the concept of the article better.

Very few things that I suggest to change are the following:

I suggest to include a column in the tables 3 and 4 with the radiological diagnosis.

Author: dear Reviewer, thank you for the suggestion. We added a column with CT and MRI diagnosis in tables 3 and 4 respectively.

Lines 99-103: Please remove the detailed description of neurological examination in the M&M section. The description of neurological examination is provided in separate dedicated textbooks.

Author: thank you for your suggestion. We agree with you and we think it can be redundant. However, since the article is not addressed only to neurologists, we prefer to remind the reader what constitutes a neurological examination.

If you prefer to keep the description of the neurological examination in the article, please add a table describing the neurological examination rather than describing it in a text.

Author: thank you for the suggestion. We delated the description of the neurological examination in the M&M section. Furthermore, we specify that a detailed description of the neurological examination can be found on Supplementary Table 1.

Lines 129-130: Please describe who were reviewing the CT and MR images - were they in all instances evaluated and reported by a board certified diagnostic imaging specialists?

Author: thank you for the question. We confirm that all CT scans and MRI studies were revised by or Head of Department of Diagnostic Imaging or by board certified diagnostic imaging specialists.

What is the degree/undergone training/affiliations (board certified?) of the head of the DI department? Please include a more detailed description in the text. Please add the specification if the diagnostic imaging specialists interpreting the images were board certified. The same applies to neurologists involved in the study.

Author: thank you for the suggestion. We specified that our Head of the Diagnostic Imaging department is a DVM, PhD (Radiology) with 20-year experience in advanced imaging (GB), while the other diagnostic imaging specialist is DVM with a second level master in diagnostic imaging (MSC Imaging). Neurologists involved in the study are a board certified neurologist (Dipl.ECVN) with a PhD in Neurology (MM) and a Neurology resident (GG).

Table 1: please specify in the table section long-term follow-up what was the time period of follow-up in each case (e.g., 7 months etc.)

Author: thank you for the suggestion. Long-term follow up was recorded only when available using medical records and phone calls to owners. Unfortunately, due to the retrospective nature of this study, we did not set a minimum long-term follow-up time. However, the lower long-term follow-up time was 1 month.

Please include in the table section describing the long-term follow-up in cases where it says the animal was ‘alive’ specification for how long after the diagnosis the animal survived until the study was performed.

Author: thank you for the suggestion. This information was added in Table 5.

Reviewer 2 Report

Comments to the authors

Thank you for addressing some my previous comments. However, some changes are still needed.

Abstract

Clinical (neurological) signs and neurological examination are not the same therefore I would strongly recommend to always be discussed individually.

Line 12 – The aim of the study is also to describe the abnormal neurological signs…

Please delete percentages as only 19 dogs.

Material and methods

Please amend - This is an observational retrospective study. The study population included client-owned dogs with presumed PA based on the MRI and/or CT examination and neurological abnormalities, either in the clinical history or the neurological examination.

Despite your efforts I am afraid it is not possible to suspected PA only based on the clinical signs or the neurological examination, unless a pituitary mass is already known which was not the case for every case.

Inclusion criteria

Could the authors explain why only 2 cases were included?

Bertolini, G.; Rossetti, E.; Caldin, M. Pituitary apoplexy-like disease in 4 dogs. J. Vet. Intern. Med. 2007, 21, 1251–1257.
Briola, C.; Galli, G.; Menchetti, M.; Caldin, M.; Bertolini, G. Pituitary tumour apoplexy due to pituitary adenoma in a dog: Clinical, 3T MRI and CT features. Vet. Rec. Case Reports 2020, 8.

Case population

I understand one case was excluded but it would have been interesting to discuss that some dogs with PA might still be clinically normal.

Neurological findings

Clinical (neurological) signs and neurological examination are not the same therefore I would strongly recommend to always be discussed individually.

Please amend – table 2

Line 189 – behavioural alterations (e.g aggression) is not part of the neurological examination but belongs to the clinical history

Figure 3 - How would the authors explain the balance loss in case 2 based on those imaging findings? I would recommend to use a more straightforward images to help the readers understand.

Table 3 – please delete detailed

Table 4 – please delete detailed

Discussion

Line 394 – use accurate instead of certain

Line 402-406 - …..all included cases had enlarged (please write large) pituitary masses (P:B ratio > 0.31) – did the Dog #2 (Figure 3) have a large pituitary mass? Please explain

Line 458 – please amend – identifying instead of identified

Conclusion

Please amend - The present study represents a case series of dogs with a presumed diagnosis of PA based on imaging findings.

Author Response

REV 2

Dear Reviewer,

The authors would like to thank you once again for the useful suggestions.

Inclusion criteria in the M&M section were better specified. Results were more clearly described and examined in the discussion section.

Finally, the paper has been extensively revised by a native English speaker in order to improve its form.

We hope you will appreciate changes made so that the article can now be considered for publication.

Comments to the authors

Thank you for addressing some my previous comments. However, some changes are still needed.

Abstract

Clinical (neurological) signs and neurological examination are not the same therefore I would strongly recommend to always be discussed individually.

Author: dear Reviewer, thank you for the suggestion. We separated the description of behavioural abnormalities from the description of the other neurological signs in the abstract.

Line 12 – The aim of the study is also to describe the abnormal neurological signs…

Author: thank you for the suggestion, we changed it.

Please delete percentages as only 19 dogs.

Author: thank you for the suggestion, we delated percentages.

Material and methods

Please amend - This is an observational retrospective study. The study population included client-owned dogs with presumed PA based on the MRI and/or CT examination and neurological abnormalities, either in the clinical history or the neurological examination. Despite your efforts I am afraid it is not possible to suspected PA only based on the clinical signs or the neurological examination, unless a pituitary mass is already known which was not the case for every case.

Author: dear Reviewer, thank you for the suggestion. Probably there was a misunderstanding. Our study population was retrospectively selected using MRI/CT findings (suggestive of PA) of dogs that undergone a complete neurological examination within 24-48 hours, so that we could compare the onset, neurological signs and outcome (when available) in dogs with diagnosis of PA (that have imaging confirmation). Therefore, we did not base our PA diagnosis on the clinical or neurological examination, as we both agree that this would be impossible to do. Hoping that this has been clarified, we changes a bit the paragraph in mat&met, in order to make it more easily to understand.

Inclusion criteria

Could the authors explain why only 2 cases were included?

Bertolini, G.; Rossetti, E.; Caldin, M. Pituitary apoplexy-like disease in 4 dogs. J. Vet. Intern. Med. 200721, 1251–1257.
Briola, C.; Galli, G.; Menchetti, M.; Caldin, M.; Bertolini, G. Pituitary tumour apoplexy due to pituitary adenoma in a dog: Clinical, 3T MRI and CT features. Vet. Rec. Case Reports 20208.

Author: thank you for the question. The remaining three cases were not included in the study because they date back prior to the inclusion time period used for this study.

Case population

I understand one case was excluded but it would have been interesting to discuss that some dogs with PA might still be clinically normal.

Author: thank you for the suggestion. We agree that the description of diagnostic findings suggestive of PA in clinically normal dogs would have been interesting. However, this specific case was excluded both for the absence of clinical and neurological signs, and because imaging findings were not suggestive of acute haemorrhages. Hence, the clinical and neurological examination would not have represented a realistic description of signs during or immediately after a PA. We specified this information in the text. We also think this is a study limitation, as our interest in select the study population with restrictive inclusion criterias would have left dogs with absence of neurological signs. Hence, we added this aspect in the discussion.

Neurological findings

Clinical (neurological) signs and neurological examination are not the same therefore I would strongly recommend to always be discussed individually.

Author: thank you for the suggestion, we definetly agree with you. Therefore, we specified and discussed the different findings (onset and clinical/neurological signs at onset vs neurological examination findings) dividing the two paragraphs (3.2 Onset and 3.3 Neurological findings). The authors hope that this division, helped by the presence of tables and supplemetray table, allows a better understanding of the two entities.

Please amend – table 2

Author: see the comment above.

Line 189 – behavioural alterations (e.g aggression) is not part of the neurological examination but belongs to the clinical history

Author: dear Reviewer, we thank you for the suggestion. However, the evaluation of sensorium and mental attitude is definetly a part of the neurological examination. As the Prof De Lahunta states in his book (Veterinary Neuroanatomy and Clinical Neurology; V ed; chapter 21 The Neurological Examination, pp 535) “An assessment should be made and recorded of the patient’s sensorium, its mental attitude, and response to the immediate environment and attitude to being handled by you. (…) Descriptive terms for this portion of the examination include alert and responsive, depressed, lethargic, obtunded, semicoma (stupor), and coma. (…) Other descriptions include acting vaguely responsive, disoriented, hyperactive, propulsive, and aggressive.” If the Reviewer think that the term “behaviour” is inappropriate, the Authors suggest to change it with “mental attitude”. Otherwise, as described elsewhere, the Authors would be very thankful to leave it as “behaviour” assessment.

Figure 3 - How would the authors explain the balance loss in case 2 based on those imaging findings? I would recommend to use a more straightforward images to help the readers understand.

Authors: thank you for the suggestion. Revising the whole paper, we agree with you that this image could be confusing and we preferred to add a new one (Dog#1).

Table 3 – please delete detailed

Author: thank you, we delated it.

Table 4 – please delete detailed

Author: thank you, we delated it.

Discussion

Line 394 – use accurate instead of certain

Author: thank you, we changed it.

Line 402-406 - …..all included cases had enlarged (please write large) pituitary masses (P:B ratio > 0.31) – did the Dog #2 (Figure 3) have a large pituitary mass? Please explain

Author: thank you, we changed it. We confirm that dog#2 had a large pituitary mass with a P:B ratio of 0.35. However, as we understand this case is a little tricky, we changed it.

Line 458 – please amend – identifying instead of identified

Author: thank you, we changed it.

Conclusion

Please amend - The present study represents a case series of dogs with a presumed diagnosis of PA based on imaging findings.

Author: thank you, we changed it.